# SEE THE TEXT: FROM TOKENIZATION TO VISUAL READING

## ABSTRACT

People see text. Humans read by recognizing words as visual objects, including their shapes, layouts, and patterns, before connecting them to meaning, which enables us to handle typos, distorted fonts, and various scripts effectively. Modern large language models (LLMs), however, rely on subword tokenization, fragmenting text into pieces from a fixed vocabulary. While effective for high-resource languages, this approach over-segments low-resource languages, yielding long, linguistically meaningless sequences and inflating computation. In this work, we challenge this entrenched paradigm and move toward a vision-centric alternative. Our method, SEETOK, renders text as images (visual-text) and leverages pretrained multimodal LLMs to interpret them, reusing strong OCR and text–vision alignment abilities learned from large-scale multimodal training. Across three different language tasks, SEETOK matches or surpasses subword tokenizers while requiring 4.43× fewer tokens and reducing FLOPs by 70.5%, with additional gains in cross-lingual generalization, robustness to typographic noise, and linguistic hierarchy. SEETOK signals a shift from symbolic tokenization to human-like visual reading, and takes a step toward more natural and cognitively inspired language models.

## 1 INTRODUCTION

*Huamn mnid deos not raed ervey lteter by istlef, but the wrod as a wlohes* (Rawlinson, 1976).
*– Graham Rawlinson*

Even with internal letters scrambled, humans can reconstruct the intended words with remarkable ease. The striking phenomenon, commonly referred to as *typoglycemia* (Johnson et al., 2007), highlights the profound robustness of human reading. Psychologists found that this ability is rooted in the Visual Word Form Area (VWFA), a brain region that identifies familiar words from visual word shapes (Dehaene & Cohen, 2011; McCandliss et al., 2003; Wimmer et al., 2016). Scrambled words typically preserve their overall shape and salient letter features, which allows the VWFA to tolerate noisy inputs and recover the intended words (Rayner et al., 2012; Agrawal et al., 2020). By leveraging holistic visual patterns and morphological cues, humans not only read efficiently and maintain robustness against noisy text (Wang et al., 2024b), but can also acquire multiple languages and writing systems with remarkable flexibility (Cohen et al., 2002; Dehaene, 2010).

In contrast, modern LLMs (Bai et al., 2025; Zhang et al., 2024a) follow a strikingly different path, leaning heavily on subword tokenization techniques, such as Byte-level BPE (Wang et al., 2020), which break text into discrete subword units from a fixed vocabulary, shaping a unique narrative of how machines process language. While effective for high-resource languages like English, this approach discards the continuous visual and morphological cues inherent in written languages. This makes tokenization highly sensitive to typos and minor perturbations (Chai et al., 2024b), which can significantly disrupt token sequences, with no ability to leverage visual similarity for correction. In multilingual contexts, it forces a compromise between inadequate coverage for low-resource languages and impractically large vocabularies (Rust et al., 2022).

We rethink the entrenched subword tokenization in LLMs and turn to a more *human-like* approach. The human brain is highly plastic, leveraging a shared **visual–linguistic pathway** across languages to map word shapes onto meanings seamlessly, as shown in Figure 1 (left). Inspired by this mechanism, we introduce SEETOK, a simple yet powerful vision-centric tokenization method for LLMs. Specifically, SEETOK first renders text into images and leverages the visual encoders of pretrained MLLMs

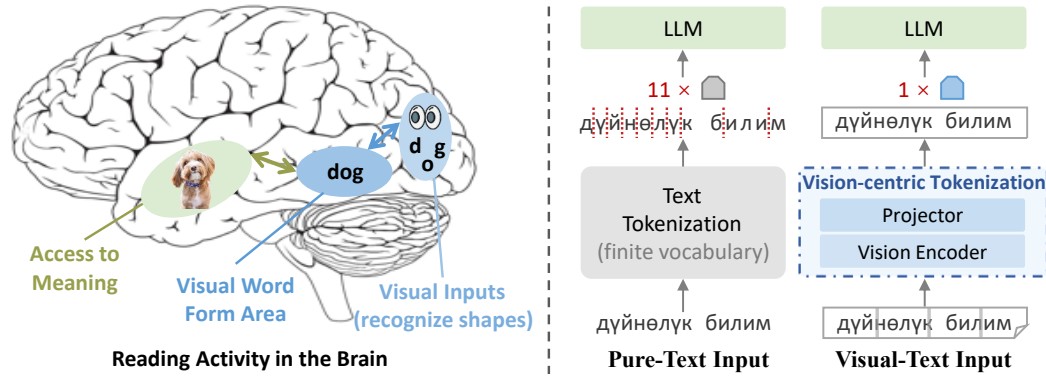

**Figure 1: Left:** Reading proceeds through a **visual–linguistic pathway**: the **visual** stream identifies letter shapes and patterns in the visual cortex and packages them into recognizable word forms via the visual word form area; the **linguistic** stream in the left-hemisphere derives meaning. **Right:** Subword tokenization tends to *over-segment low-resource languages* due to insufficient vocabulary coverage, *e.g.*, a 2-word Kyrgyz phrase ("*world knowledge*") is split into 11 text tokens. Our vision-centric tokenization instead compresses the phrase into a single visual token by aggregating features from four adjacent image patches through the projector.

(*e.g.*, Qwen2.5-VL (Bai et al., 2025)) to extract textual representations, which are then passed to the LLM backbone for deeper processing. Benefiting from large-scale vision-language pretraining, these visual encoders naturally exhibit strong OCR ability and robust text–vision alignment (Yuan et al., 2025; Lin et al., 2025; Liu et al., 2024b), making them a promising alternative to conventional text tokenization. To enhance instruction-following in the visual modality, we introduce vision-centric instruction tuning, where instruction texts are rendered as images (*i.e.*, visual-text instructions) and the MLLM is adapted with lightweight LoRA (Hu et al., 2022) layers. This simple yet effective procedure enables MLLMs to interpret visual-text instructions on par with pure-text ones (*cf.* Figure 6), without costly training from scratch or architectural modifications.

We primarily evaluate our SEETOK on the widely-used open-source models JanusPro (Chen et al., 2025b) and Qwen2.5-VL (Bai et al., 2025). Across three representative natural language understanding tasks, SEETOK achieves performance on par with text-tokenization baseline, while requiring **4.43×** fewer visual tokens and reducing FLOPs by **70.5%**. In multilingual translation covering 13 languages, SEETOK further shows stronger cross-lingual transfer compared to the text-tokenization counterpart, achieving **86%** lower fertility (*i.e.*, fewer tokens per word) and a **+3.87** gain in COMET-22 scores. Moreover, SEETOK exhibits strong compositionality and robustness to input perturbations, showing substantially smaller performance drops than the text-tokenization model across character-level, word-level, and visual-level attacks. Importantly, SEETOK generalizes well to other MLLMs, including JanusPro 1B (Chen et al., 2025b) and Qwen2.5-VL 7B (Bai et al., 2025).

Below, we summarize the advantages of our vision-centric tokenization, highlighting that representing text visually is a promising and valuable direction for future research. ❶ **Efficiency.** Compared to text tokenization, our vision-centric tokenization significantly reduces token counts *across 14 diverse languages* (*cf.* Table 7), with even greater benefits for low-resource languages (*e.g.*, 4.43× for English, 13.05× for Georgian). This advantage arises from its language-agnostic design, avoiding the inherent bias of text tokenization toward high-resource languages (Truong et al., 2024). ❷ **Strong cross-lingual generalization.** Our vision-centric tokenization demonstrates robust cross-lingual generalization, achieving higher translation quality than text-tokenization counterpart for both high- and low-resource languages while avoiding excessive subword segmentation (*cf.* Sec. 4.3). ❸ **Robustness to orthographic perturbations.** Our vision-centric tokenization is less sensitive to input perturbations than conventional text tokenization (*cf.* Sec. 4.4). By processing text as continuous visual patterns, minor edits (*e.g.*, insertions, deletions, substitutions) affect only local features, while the overall word shape remains intact, resulting in robust representations. ❹ **Hierarchical structure awareness.** Subword tokenization splits words into discrete and independent units without explicitly modeling the hierarchy from characters to words (Chai et al., 2024a). In contrast, our vision tokenization can naturally learn linguistic structural regularities from the holistic visual patterns of text, resulting in stronger compositionality (*cf.* Figure 5) (Peng et al., 2025).

## 2 RELATED WORK

**Text Tokenization.** Text tokenization (Kenton & Toutanova, 2019; Kudo & Richardson, 2018; Sennrich et al., 2016a) is the first step in natural language processing, segmenting the strings of text into smaller units. Based on the granularity of segmentation, tokenization can be broadly classified into three types. 1) *Character-level* tokenization treats each character or byte as an atomic token (Xue et al., 2022). This design keeps the vocabulary small, but results in long input sequences that substantially increase memory and computation costs. Several strategies (Yu et al., 2023; Pagnoni et al., 2024) have been developed to mitigate this limitation. 2) *Word-level* tokenization operates on entire lexical items, typically segmented by whitespace or language-specific heuristics (Bengio et al., 2003). They are efficient for frequent words, but face out-of-vocabulary (OOV) issues and demand huge vocabularies in multilingual settings, which inflate memory usage and make the softmax in the output layer computationally expensive. 3) *Subword-level* tokenization, such as BPE (Sennrich et al., 2016b), WordPiece (Devlin et al., 2019), and Unigram (Kudo, 2018), segment words into subword units and are now widely used. They balance vocabulary size and coverage while mitigating OOV issues, but break morphological boundaries and are sensitive to surface noise (Rust et al., 2022). In multilingual contexts, the *fixed* vocabulary is *primarily allocated to high-resource languages*, leaving low-resource languages with limited coverage. Consequently, words in *low-resource languages are over-segmented*, sometimes almost at *character-level*, leading to significantly longer token length. In this work, we explore a vision-centric tokenization route that treats raw text as images. This method promotes multilingual fairness, achieving *low token fertility even for low-resource languages*.

**Vision-centric Method.** Subword tokenization (Wang et al., 2020), though effective, suffer from vocabulary bottlenecks (Rust et al., 2022), noise sensitivity (Chai et al., 2024a), and multilingual unfairness (Limisiewicz et al., 2023). An emerging line of work (Gao et al., 2024; Lotz et al., 2023; Zhuang et al., 2025; Zhang et al., 2024b) circumvents this limitation by processing text as images. A representative method, PIXEL (Rust et al., 2022), applies a VIT-MAE (He et al., 2022) with masked patch prediction on rendered text images, achieving strong multilingual performance and robustness to noise. Follow-up work extends this paradigm by addressing input redundancy (Lotz et al., 2025) and exploring alternative objectives such as next-patch prediction (Tai et al., 2024; Chai et al., 2024b) and patch-and-text prediction (Gao et al., 2024). PIXAR (Tai et al., 2024) provides a compelling demonstration of a fully vision-centric design, generating text autoregressively through image patches. However, its generation capability is restricted to short pixel-based text sequences. PTP (Gao et al., 2024) explores both encoder-only and decoder-only variants, where the decoder-only setup is similar to Fuyu-style (Bavishi et al., 2023) that do not include image encoder. Despite improvements, these methods are mainly evaluated on relatively simple NLP benchmarks (Wang et al., 2018) and consistently lag behind text-only baseline (Devlin et al., 2019). Other efforts exploit rendered text images to increase the effective context length for LLMs (Xing et al., 2025) and MLLMs (Wang et al., 2024a). In multimodal learning, CLIPPO (Tschannen et al., 2023), a single vision transformer model, unifies image and text processing by treating text visually. It achieves performance comparable to CLIP-style models (Radford et al., 2021) with half the number of parameters on image classification and text/image retrieval tasks. Pix2Struct (Lee et al., 2023) learns to parse masked webpage screenshots into simplified HTML, improving visually situated language understanding. Recent studies (Wang & Ma; Baek et al., 2025) provide valuable insights into how existing multimodal models internally process text in visual inputs. Wang & Ma show that early layers mainly rely on visual texture, with semantic understanding emerging only in the final blocks. Baek et al. (2025) identify specialized "OCR heads" within MLLMs that are responsible for reading text directly from images. These findings complement the growing interest in treating text visually. Rather than training from scratch, our SEETOK builds on pretrained MLLMs with inherent OCR capacity, enabling more effective handling of visual-text instructions. Compared to subword text tokenization, SEETOK achieves greater robustness to surface perturbations, stronger cross-lingual generalization, and deeper capture of hierarchical structure in language.

## 3 METHODOLOGY

SEETOK proposes a novel approach in which text is not fed as discrete tokens but rendered into images, enabling the model to perceive and process textual content visually (visual-text).

## 3.1 OVERALL PIPELINE

Figure 2 illustrates the overall pipeline of SEETOK. Given an input text sequence, we first apply a **visual renderer** that transforms the raw string into a rendered text image. The image is then processed by the **vision-centric tokenization** (*i.e.*, vision encoder and MLP projector from MLLMs), which substitutes for standard text tokenization and empowers the LLM to perceive text directly in visual form rather than as discrete tokens. The LLM subsequently consumes these encoded visual features to perform downstream reasoning and generation. We primarily base our study on widely used Qwen2.5-VL 3B, Qwen2.5-VL 7B (Bai et al., 2025), and JanusPro (Chen et al., 2025b).

Although modern MLLMs possess strong OCR and vision–language alignment (Yuan et al., 2025; Lin et al., 2025), they are rarely exposed to *visual-text instructions* (*i.e.*, instructions presented as rendered images) during pretraining. This results in a distribution gap, causing weaker visual-text instruction-following ability compared to pure-text instructions. To close the gap, we integrate LoRA adapters (Hu et al., 2022) into both the vision encoder and the LLM. LoRA adapters improve fine-grained text perception on the vision encoder side and align instruction-following on the LLM side, enabling SEETOK to handle visual-text prompts effectively with negligible training overhead compared to pretraining from scratch, while incurring no additional inference parameters.

## 3.2 VISUAL TEXT TOKENIZATION

**Visual Renderer.** The core component of SEETOK is a visual renderer that transforms raw textual data into RGB images $\mathcal{X}_{\text{img}} = \{x_m \in \mathbb{R}^{H \times W \times C}\}_{m=1}^{M}$, where $M$ denotes the number of rendered text images and can be dynamically adjusted based on the length of the input text.

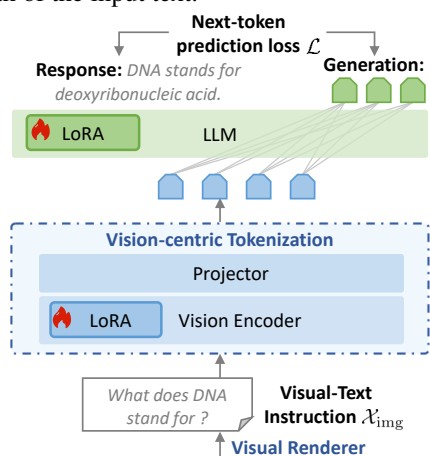

If the vision encoder supports variable resolutions (*e.g.*, the 3B and 7B versions of Qwen2.5-VL (Bai et al., 2025)), both the image height $H$ and width $W$ can be scaled to match the text length. In our training setup, we configure $M = 1$ with height $H = 14$, width $W = 3584$, and $C = 3$ channels, corresponding to an image of resolution $224 \times 224$. Text is rendered using the *Google Noto Sans* typeface with a font size of 7px.

**Vision-centric Tokenization.** The visual-text is first processed by the vision encoder to extract patch-level features. On average, a $14 \times 14$ image patch encodes roughly 1.1 Qwen tokens in English, highlighting the compactness of the visual representation. A two-layer MLP projector then aggregates four neighboring patches and projects them into a dimension aligned with the text embeddings of the LLM, reducing the token sequence length by $4\times$. Together, the vision encoder and projector function as a *"visual" text tokenization*, providing an efficient and effective substitute for standard text tokenization. In contrast to *subword vocabulary biased toward high-resource languages*, patch-

Figure 2: **Overview of SEETOK.** Text is rendered into an image, processed by the vision-centric tokenization, and fed to the LLM. LoRA layers further boost its ability to follow visual-text instructions.

based segmentation ensures that diverse languages are encoded fairly **without requiring vocabulary enlargement**. This design yields substantial efficiency gains compared to text tokenization, reducing fertility (*i.e.*, the average token count per word) by **86%** on average across 13 languages, including both high- and low-resource languages (*cf.* Sec. 4.3).

## 3.3 VISION-CENTRIC FINETUNING

Pretrained MLLMs demonstrate strong OCR capabilities and excel at recognizing textual content within images (Yuan et al., 2025; Lin et al., 2025; Liu et al., 2024b). However, when instructions are provided as visual-text instead of pure-text, model performance drops significantly (*cf.* Table 1). This indicates that, although the model can accurately read the text, it struggles to interpret it as an instruction and perform reasoning accordingly. This gap may be because the MLLMs are rarely exposed to visual-text instructions during pretraining, and thus fail to associate visualized text with the same instruction-following semantics as conventional text tokens.

Table 1: Our vision tokenization-based SEETOK significantly enhances Qwen2.5-VL 3B with visual-text input on diversity language understanding tasks. On average across multiple types of language tasks, SEETOK matches the performance of the text-tokenization baseline Qwen2.5-VL 3B with pure-text input.

| Models | Text Source | TriviaQA | NQ | PopQA | MMLU | SST5 | Avg. |
|---|---|---|---|---|---|---|---|
| Qwen2.5-VL 3B | Pure-Text | 41.92 | 29.31 | 24.64 | 61.91 | 28.80 | 37.32 |
| Qwen2.5-VL 3B | Visual-Text | 37.55 | 21.13 | 20.16 | 32.31 | 25.21 | 27.27 |
| + SEETOK | Visual-Text | **43.53**(5.98↑) | 24.14(**3.01**↑) | 24.26(**4.10**↑) | 52.52(**20.21**↑) | **44.40**(**19.19**↑) | **37.77**(**10.50**↑) |

To address this limitation, we perform instruction tuning using LoRA layers (Hu et al., 2022) applied to both the vision encoder and the LLM. During tuning, instructions are rendered as text images, while target answers remain in textual form to compute the next-token prediction loss. Formally, given an instruction $I$ rendered as images $\mathcal{X}_{\mathrm{img}}$ and a target response sequence $\mathbf{y} = (y_1, \ldots, y_T)$, we optimize the standard autoregressive generation loss:

$$\mathcal{L} = -\sum_{t=1}^{T} \log P\big(y_t \mid y_{<t}, \mathcal{X}_{\mathrm{img}}\big), \tag{1}$$

where $\mathcal{X}_{\mathrm{img}}$ is encoded via the vision encoder and MLP to serve as the instruction signal, and the decoder LLM generates the answer token by token. This training explicitly encourages the model to interpret visualized instructions correctly and generate responses that align with textual answers. By leveraging pretrained MLLMs, SEETOK realizes a vision-centric tokenization in a lightweight and efficient manner, **eliminating the need to train from scratch**. Crucially, our experiments indicate that *keeping the projector frozen* is essential for stable performance during instruction tuning (*cf.* Table 6), as it preserves the robust cross-modal alignment learned from large-scale pretraining.

## 4 EXPERIMENT

### 4.1 EXPERIMENTAL SETUP

**Datasets and Implementation Details.** We validate SEETOK with Qwen2.5-VL 3B, Qwen2.5-VL 7B (Bai et al., 2025), and JanusPro (Chen et al., 2025b). To reduce computational overhead, we employ DeepSpeed with ZeRO stage-2 (Rasley et al., 2020) and float16 precision. Full hyperparameter details are provided in Appendix A. We employ OpenHermes 2.5 (Teknium, 2023) as the instruction-tuning corpus, providing a larger-scale and high-quality collection of diverse instruction–chat samples. Due to resource limitations, we exclude excessively long samples to prevent out-of-memory issues, resulting in a filtered corpus of 658k instances.

**Downstream Evaluation.** To comprehensively compare SEETOK with the text-tokenization counterpart, we evaluate both on natural language understanding and multilingual translation benchmarks, analyzing cross-lingual transfer, multilingual efficiency, compositionality, and robustness to noise.

### 4.2 EVALUATION ACROSS MULTIPLE LANGUAGE UNDERSTANDING TASKS

To assess the effectiveness of our SEETOK, we evaluate on multiple representative natural language understanding tasks, spanning open-domain question answering (TriviaQA (Joshi et al., 2017), NQ (Kwiatkowski et al., 2019), and PopQA (Mallen et al., 2023)), general knowledge reasoning (MMLU (Hendrycks et al., 2021)), and sentiment classification (SST5 (Socher et al., 2013)). We report Exact Match (EM) for QA and accuracy for MMLU and SST5.

**Effectiveness.** As shown in Table 1, SEETOK (vision-centric tokenization over visual-text inputs) **matches or even surpasses** the text-tokenization counterpart (Qwen2.5-VL 3B), averaging 37.77 compared to 37.32 across five datasets. Notably, SEETOK outperforms the text-tokenization baseline on TriviaQA (+1.61) and SST5 (+15.60). These two tasks rely heavily on surface-form cues such as spelling, capitalization, and negation. Subword text tokenization

Table 2: **Evaluating efficiency between standard text tokenization and vision tokenization** on the TriviaQA dataset (Joshi et al., 2017) based on SEE-TOK. Compression ratio $\Delta$ is the ratio of the text-token count to the number of visual-text tokens.

| Text Source | $\Delta$ | Latency | TFLOPs |
|---|---|---|---|
| Pure-Text | - | 5.02 | 3.12 |
| Visual-Text | 4.43 | **3.34** | **0.92** |

Table 3: Translation performance from high-resource languages to English. Fertility (FET) measures the average number of tokens used to represent a single word. COMET-22 score (COM) evaluates overall translation quality. † denotes the same LoRA setup as SEETOK, with pure-text training input.

| Models | Text Source | de | | cs | | is | | zh | | ru | | Avg. | |
|---|---|---|---|---|---|---|---|---|---|---|---|---|---|
| | | COM↑ | FET↓ | COM↑ | FET↓ | COM↑ | FET↓ | COM↑ | FET↓ | COM↑ | FET↓ | COM↑ | FET↓ |
| Qwen2.5-VL 3B | Pure-Text | 67.25 | 1.89 | 62.02 | 2.81 | 53.63 | 2.71 | 57.51 | 1.09 | 63.16 | 2.53 | 60.71 | 2.21 |
| Qwen2.5-VL 3B† | Pure-Text | **67.88** | 1.89 | 62.05 | 2.81 | 53.89 | 2.71 | 58.12 | 1.09 | 65.33 | 2.53 | 61.45 | 2.21 |
| Qwen2.5-VL 3B | Visual-Text | 47.49 | **0.42** | 41.02 | **0.38** | 34.37 | **0.37** | 46.77 | **0.21** | 46.44 | **0.49** | 33.72 | **0.37** |
| + SEETOK | Visual-Text | 65.63 | **0.42** | **64.89** | **0.38** | **54.97** | **0.37** | **68.94** | **0.21** | **71.42** | **0.49** | **65.17** | **0.37** |

often fragments or obscures such information, particularly for rare words and entities (Tanaka et al., 2021; Truong et al., 2024). In contrast, the vision-centric tokenization preserves character-level fidelity, enabling the model to capture these signals more faithfully. MMLU (Hendrycks et al., 2021) is a knowledge-intensive benchmark spanning multiple domains, formulas, and logical reasoning, which relies more heavily on world knowledge learned from large-scale textual pretraining. Since the vision pathway has not been exposed to comparable amounts of such data, a performance gap remains. Similar pretraining conducted on visual-text could potentially further narrow this gap, a trend already reflected in the scaling experiments (see Appendix E).

**Efficiency.** Vision-centric tokenization provides substantial efficiency benefits. We quantify efficiency on TriviaQA (Joshi et al., 2017), comparing two tokenization schemes: standard text tokenization and vision-centric tokenization. Both models are based on SEETOK. We report the compression ratio $\Delta$ defined as the dataset-level average text tokens divided by the average visual-text tokens, along with FLOPs and end-to-end latency (in seconds). As summarized in Table 2, SEETOK with visual-text input achieves **4.43× reduction in token length**, along with **70.5% lower FLOPs** and **33.5% faster latency** compared to the model with pure text input, while **maintaining comparable performance**. These efficiency gains make vision-centric tokenization particularly attractive for resource-constrained environments, where reducing inference cost is critical. Latency measures the total wall-clock time from input reception to the generation of 64 output tokens. More detailed analysis of the memory and computational costs in Appendix F

## 4.3 MLTILINGUAL TRANSLATION EVALUATION

To evaluate the multilingual capabilities of our approach, we test the translation performance across multiple languages, divided into two groups: **i) High-resource Languages:** de (German), cs (Czech), is (Icelandic), zh (Chinese), ru (Russian). **ii) Low-resource Languages:** ky (Kyrgyz), uz (Uzbek), ka (Georgian), lt (Lithuanian), lv (Latvian), bg (Bulgarian), mk (Macedonian), mg (Malagasy). We report the **COMET-22 score (COM)** for the translation from each of these languages to English (Rei et al., 2022), as suggested by Freitag et al. (2023). A higher COM indicates better translation quality in terms of fluency and correctness. We also calculate **Fertility (FET)**, a metric for assessing tokenization performance (Rust et al., 2021), defined as the average number of tokens per word. For word segmentation, we use Jieba for zh and whitespace splitting for other languages (Ali et al., 2024).

**High-resource Languages.** Since the model Qwen2.5-VL 3B has not encountered multilingual visual-text instructions, we finetune it on ALMA (Xu et al., 2024), a small but high-quality bilingual corpus, to enable effective multilingual instruction following in the visual-text form. To ensure fair comparison, Qwen2.5-VL 3B is also finetuned on the same dataset ALMA (Xu et al., 2024) with pure-text input, denoted as Qwen2.5-VL 3B†. Further training details are provided in the Appendix B. Following ALMA (Xu et al., 2024), we test on WMT22 test data (Freitag et al., 2022), except for Icelandic (is), which is tested on WMT21 (Freitag et al., 2021). Table 3 illustrates that SEETOK enhances the performance of Qwen2.5-VL with visual-text input, achieving an average COM improvement of **+31.45** across five languages. SEETOK with visual-text input also outperforms the text tokenization baseline Qwen2.5-VL 3B†, particularly for **non-Latin languages** such as zh and ru. This suggests that the vision-centric tokenization *offers a stronger advantage for languages that differ more from English in terms of grammar and morphology*.

**Low-resource Languages.** Since it is unclear whether Qwen2.5-VL 3B has seen these low-resource languages during pretraining, we finetune it using two methods: (i) pure text finetuning and (ii) visual-text finetuning. The training parallel data is provided by X-ALMA (Xu et al., 2025), and we use the FLORES test set (Costa-Jussà et al., 2022) for evaluation. As shown in Figure 3 (right), finetuning with visual-text input results in a **higher average COM** (58.12) compared to pure text finetuning

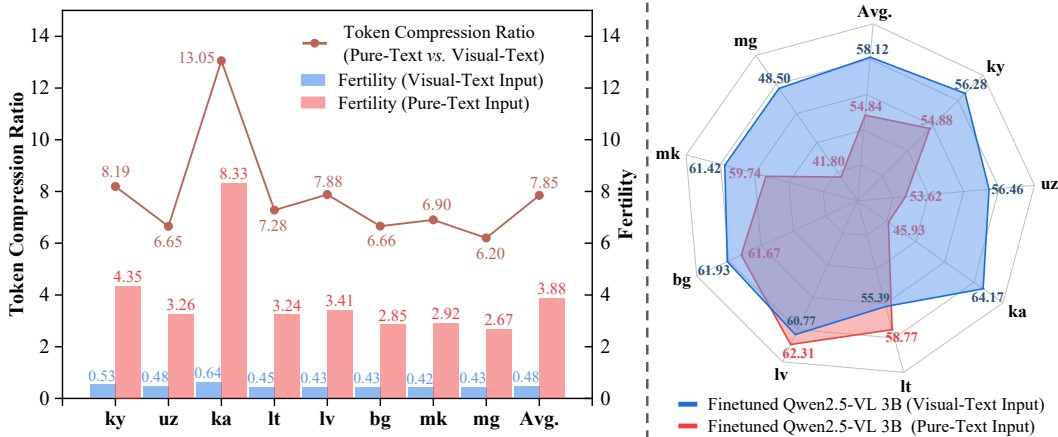

Figure 3: **Left:** Fertility and token compression ratio across low-resource languages, comparing text and vision-centric tokenization. **Right:** COMET-22 scores on FLORES for translations from low-resource languages into English, comparing Qwen2.5-VL 3B trained with visual-text input and with pure-text input.

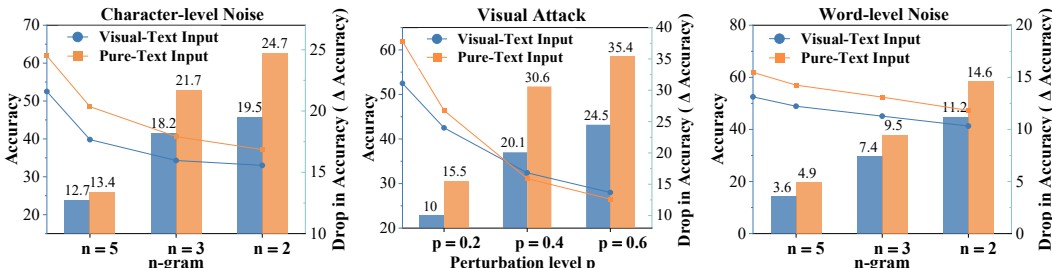

Figure 4: **Accuracy drop** on MMLU (Hendrycks et al., 2021) under different orthographic perturbations (character-, visual-, and word-level noise). The vision tokenization–based model (blue) shows **markedly smaller declines** than the text-tokenization counterpart (orange), demonstrating stronger robustness to surface noise.

(54.84). Higher fertility observed in pure-text input (Figure 3, left) corresponds to over-segmentation, which hampers the ability of the model to learn translation patterns and decreases translation quality.

**Fertility.** In Table 3 and Figure 3 (left), the **vision-centric tokenization yields significantly lower fertility than the text tokenization across all languages**. For high-resource languages, SEETOK averages 0.37 compared to 2.21 for the text tokenization, while for low-resource languages, SEETOK averages 0.48 *vs.* 3.88. This shows that SEETOK *is more efficiently and treats all languages fairly*. In contrast, the text tokenization favors high-resource languages like English but excessively fragments low-resource languages, even down to the *character level* (*e.g.*, 8.33 fertility for ka). Furthermore, the vision-centric tokenization substantially **reduces token length**, achieving an average compression ratio of 7.85 for low-resource languages (Figure 3, left) and 5.71 for high-resource languages, relative to standard text tokenization. Full results can be found in the Appendix G.

## 4.4 FINE-GRAINED LEXICAL REASONING

**Perturbation Probing.** We assess the robustness of Qwen2.5-VL 3B with text tokenization *vs.* SEETOK with vision tokenization on MMLU (Hendrycks et al., 2021) under three perturbation types in a **zero-shot setting** (*i.e.*, without any dataset-specific fine-tuning). (i) *Character-level noise.* For low-level surface corruption, we use the TKEval-MMLU (Chai et al., 2024a), which simulates realistic typographical errors by applying within-word $n$-gram shuffling ($n \in \{2, 3, 5\}$) and random character edits such as insertions and deletions. (ii) *Visual attacks.* To evaluate perceptual robustness, we follow ECES (Eger et al., 2019), substituting Latin letters with visually similar glyphs (*e.g.*, ê for "e") at controlled perturbation levels $p \in \{0.2, 0.4, 0.6\}$. (iii) *Word-level noise.* To probe semantic robustness, words are randomly corrupted with probabilities $p \in \{0.2, 0.4, 0.6\}$, including synonym substitution and deletion. Perturbation implementation details can be found in the Appendix I.

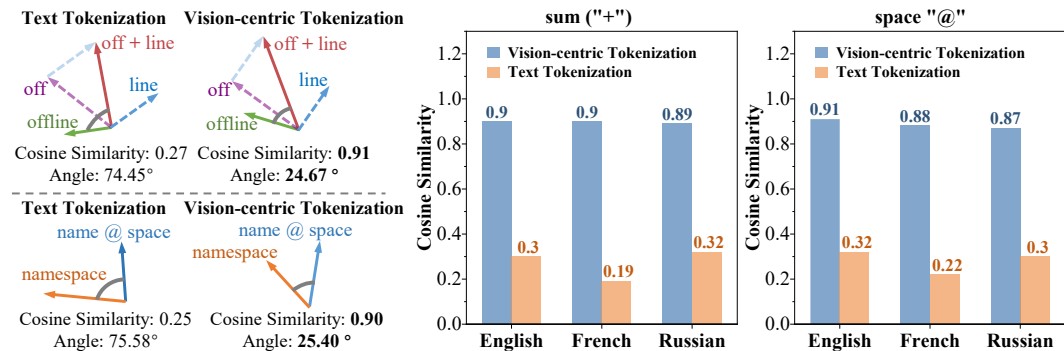

Figure 5: **Compositional evaluation of token embeddings** from text and vision tokenization across three languages. Cosine similarity and angle are computed between original full-word embedding (*e.g.*, `offline`) and its composed embedding (*e.g.*, {`off`, `line`}). **Sum ("+")** means the composed embedding is obtained by summing subword embeddings. **Space ("@")** denotes composition by concatenating subwords with a space. **Vision tokenization yields composed embeddings more consistent with the full word across all languages.**

Table 4: Evaluation under different training and inference text source settings. * indicates results on a reduced training dataset, where long samples were removed to prevent out-of-memory issues with pure-text input. ♣ denotes the same finetuning setup as SEETOK, with pure-text training input.

| Models | Training Input | Inference Input | TriviaQA | NQ | PopQA | SST5 | MMLU |
|---|---|---|---|---|---|---|---|
| Qwen2.5-VL 3B | - | Visual-Text | 37.55 | 21.13 | 20.16 | 25.21 | 32.31 |
| Qwen2.5-VL 3B | - | Pure-Text | 41.92 | 29.31 | 24.64 | 28.80 | 61.91 |
| Qwen2.5-VL 3B*♣ | Pure-Text | Pure-Text | 42.06(0.14↑) | 29.75(0.44↑) | 24.96(0.32↑) | 30.00(1.20↑) | 62.21(0.30↑) |
| + SEETOK* | Visual-Text | Visual-Text | 42.18(4.63↑) | 23.16(2.03↑) | 23.47(2.03↑) | 32.80(7.59↑) | 49.00(16.70↑) |
| + SEETOK* | Visual-Text | Pure-Text | 42.54(0.62↑) | 30.18(0.62↑) | 25.21(0.62↑) | 31.42(2.62↑) | 62.34(0.43↑) |

As illustrated in Figure 4, the vision-centric tokenization (*i.e.*, visual-text input) **suffers significantly less performance drop** than the text-tokenization (*i.e.*, pure text input) across all perturbations. Subword tokenization can drastically change token sequences even with minor input perturbations, increasing susceptibility to errors. In contrast, the vision tokenization treats characters as visual units, capturing their shape and spatial layout. Minor typographical or lexical changes affect only local visual details, leaving the overall representation largely intact and improving robustness to noise.

**Subword Compositionality.** Compositional ability enables the model to *generalize to novel combinations* instead of just memorizing patterns (Chai et al., 2024a; Peng et al., 2025). To assess the ability of text- and vision-tokenized embeddings to capture subword compositional structure, we draw on the SIGMORPHON 2022 dataset (Batsuren et al., 2022), which provides full words and their possible subword decompositions (*e.g.*, `offline` → `off, line`). As in Peng et al. (2025), we retain only full words that appear in the model vocabulary and perform experiments across English, French, and Russian. We measure cosine similarity and angle between full-word embedding and its corresponding composed embedding to evaluate compositional fidelity. The composed embeddings are constructed in two ways: (i) *sum*, by summing the embeddings of each subword, and (ii) *space*, by embedding the subwords concatenated with a space.

Figure 5 shows vision tokenization achieves cosine similarity close to 1.0 and much smaller angles than text tokenization across all languages. This suggests that **vision-based embeddings capture compositional structure far more faithfully**, as they encode each word as a sequence of visual patterns, inherently maintaining local geometric relations. By contrast, the text tokenization splits the word into independent subword units without explicitly modeling the hierarchy from characters to words. This limitation not only weakens compositional alignment but also *explains the greater sensitivity of text-tokenized embeddings to surface-level perturbations and morphological changes*.

### 4.5 ABLATION STUDY

**Extension to More LLMs.** To prove the generality of SEETOK, we test the unified model JanusPro 1B (Chen et al., 2025b), Qwen2.5-VL 7B (Bai et al., 2025) , and Llava-next-

Table 5: SEETOK consistently improves instruction-following with visual-text inputs across different model backbones. † corresponds to SEETOK on JanusPro 1B (Chen et al., 2025b), ‡ on Qwen2.5-VL 7B (Bai et al., 2023), and ⋄ on Llava-next 8B (Liu et al., 2024a).

| Models | Text Source | TriviaQA |
|---|---|---|
| JanusPro 1B | Pure-Text | 42.71 |
| JanusPro 1B | Visual-Text | 27.10 |
| + SEETOK† | Visual-Text | **35.23**(8.13↑) |
| Qwen2.5-VL 7B | Pure-Text | 58.90 |
| Qwen2.5-VL 7B | Visual-Text | 53.53 |
| + SEETOK‡ | Visual-Text | **59.65**(6.12↑) |
| Llava-next 8B | Pure-Text | 58.71 |
| Llava-next 8B | Visual-Text | 51.18 |
| + SEETOK⋄ | Visual-Text | **59.72**(8.54↑) |

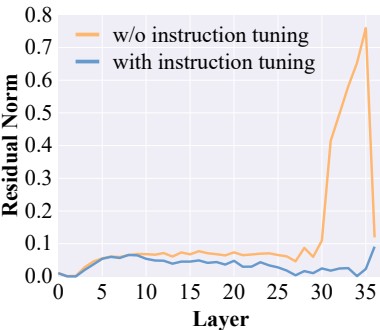

Figure 6: Layer-wise residual norms from orthogonal Procrustes alignment between visual-text and pure-text embeddings. Vision-centric instruction tuning lowers residual norm in deeper layers (31-35), reflecting more consistent processing of pure-text and visual-text inputs.

8b (Liu et al., 2024a) on TriviaQA (Joshi et al., 2017) under both pure-text and visual-text inputs, comparing performance with and w/o SEETOK. As summarized in Table 5, both backbones show degraded performance with visual-text inputs, as they have not been exposed to such instructions during pretraining. However, **integrating SEETOK yields substantial gains**, recovering or even exceeding their performance on pure-text inputs. These results confirm SEETOK consistently improves instruction-following in visual-text settings.

**Ablation on Fine-tuning Scope.** We investigate the effect of applying LoRA layers to different components of SEETOK, including the vision encoder, projector, and LLM. Table 6 shows that adapting the vision encoder and LLM while **freezing the projector** gives the best results, whereas updating the projector together with other modules degrades performance. The projector, pretrained on large-scale image–text corpora, already provides a robust alignment. Fine-tuning it on the comparatively narrow instruction-tuning data risks disrupting this alignment, leading to a drop in performance. More results are included in the Table 15 (Appendix M).

Table 6: **Ablation on fine-tuning scope.** Keeping the projector frozen is critical for stable gains, with tuning the vision encoder and LLM providing optimal performance.

| Vision Encoder | Projector | LLM | TriviaQA |
|---|---|---|---|
| | | | 37.55 |
| ✓ | ✓ | ✓ | 37.02 |
| ✓ | | ✓ | **43.53** |

**Evaluation on Vision-native Tasks.** We test SEETOK with Qwen2.5-VL 3B (Bai et al., 2023) on VQAv2 (Goyal et al., 2017), TextVQA (Singh et al., 2019), and DocVQA (Mathew et al., 2021). As shown in Table 7, its performance remains on par with Qwen2.5-VL 3B, confirming that vision-centric instruction tuning does not harm the native vision–language performance.

Table 7: SEETOK matches Qwen2.5-VL 3B on VQAv2, DocVQA, and TextVQA, showing that vision-centric instruction tuning preserves naive vision–language performance.

| Model | VQAv2 | DocVQA | TextVQA |
|---|---|---|---|
| Qwen2.5-VL 3B | 81.2 | 93.9 | 79.3 |
| SEETOK | 81.0 | 93.5 | 80.1 |

## 5 DISCUSSION

**Do Visual-Text Instruction Improvements Stem from Additional Knowledge from the New Data?** A key question is whether the gains observed after visual-text instruction finetuning arise from access to new knowledge in the finetuning corpus, or from improved ability to follow visual-text instructions. To disentangle these factors, we finetune Qwen2.5-VL 3B on the same data in two formats: visual-text and pure-text. Because pure-text input consumes substantially more tokens, we further filter samples to avoid out-of-memory issues (denoted by * in Table 4, Rows 4–6). The results reveal a striking contrast: visual-text finetuning (row 5) yields a **+16.70 improvement** over baseline (row 2), whereas pure-text finetuning offers only a marginal gain of +0.30. This indicates that the improvements primarily stem from enhanced instruction-following ability in the visual-text format, rather than access to new information. Moreover, the efficiency of visual-text tokens enables training on more examples under identical compute constraints, producing even larger gains (52.52 *vs.* 49.00).

Thus, the advantage of our approach lies not only in robustness to tokenization but also in more effective use of limited training budgets.

**Effect on Text-Only Performance after Visual-Text Instruction Tuning.** We examine how fine-tuning the model with visual-text instructions impacts its performance on pure-text inputs, using five widely recognized benchmarks for evaluation. As detailed in Table 4, Qwen2.5-VL 3B (Bai et al., 2023) finetuned on visual-text input (Row 6) achieves greater improvements on pure-text inference than the variant finetuned on pure-text data (Row 4), consistently across the five benchmarks. This improvement suggests that *even though the finetuning is performed using visual-text data, the model benefits from better cross-format generalization, enhancing its pure text performance*. The ability to process both image-based and text-based instructions likely equips the model with richer understanding capabilities that extend beyond the specific input format. Notably, finetuning with visual-text inputs is more efficient, as it uses far fewer input tokens than pure-text finetuning, allowing the model to achieve stronger improvements at lower computational cost.

**Layerwise Effect of Instruction Tuning on Cross-Modal Alignment.** A key question is whether instruction tuning helps the model treat visual-text inputs consistently with their pure-text counterparts. To probe this, we apply Orthogonal Procrustes analysis (Schönemann, 1966) on Qwen2.5-VL 3B, with 1k out-of-distribution samples from ALPAGASUS (Chen et al., 2024). This method finds the optimal linear transformation that aligns visual-text embeddings with pure-text embeddings while preserving internal geometry. We quantify alignment using the residual norm, *i.e.*, the Frobenius distance between the transformed visual-text embeddings and the corresponding pure-text embeddings. Lower residual norm indicates stronger structural similarity. Results in Figure 6 reveal that instruction-tuned models achieve progressively lower residuals in deeper layers, reflecting improved convergence between text and visual-text pathways. In contrast, the frozen model exhibits high residuals in the layers 31–35, consistent with its weaker performance on visual-text instructions. These results suggest that instruction tuning reshapes representational geometry across modalities, enabling more consistent processing of pure-text and visual-text inputs.

## 6 CONCLUSION

In this work, we introduce SEETOK, a simple yet effective vision-centric tokenization method that substitutes conventional text tokenization by encoding rendered images through pretrained vision encoders. Our approach achieves competitive or superior performance to conventional text tokenization, while offering clear advantages in multilingual efficiency, compositionality, and robustness to noise. These results highlight the promise of visual tokenization as a general alternative to prevailing subword tokenization. In future work, we plan to leverage vision encoders as a unifying interface across modalities, paving the way toward more general multimodal reasoning.

## ETHICS STATEMENT

This work does not involve human subjects, sensitive personal data, or applications with direct societal risks. All datasets used are publicly available and have been widely adopted in prior research. We therefore believe our study poses no ethical concerns beyond standard practices.

## REPRODUCIBILITY STATEMENT

To facilitate reproducibility, we provide detailed descriptions of dataset usage and model hyperparameters in Sec. 4.1 and Appendix A and B. All datasets are publicly available, and training and inference code are included in the supplemental material.

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

APPENDIX

The document is organized as follows:

- §A Experimental Details.
- §B Downstream Task Evaluation Details.
- §C Promising Direction: The Vision-centric Paradigm.
- §D Effectiveness under Limited Data.
- §E Scaling Study on Visual-Text Instruction Data.
- §F Memory and Computational Costs.
- §G Compression Ratio across High- and Low Languages.
- §H Visualization of Visual-Text and Low-resource Languages.
- §I Perturbation Implementation Details.
- §J Additional Analysis on Fairness of Perturbation Evaluation.
- §K Typoglycemia.
- §L Compositionality Across Languages.
- §M Ablation on Fine-tuning Scope.
- §N Generalization of SEETOK to Additional Tasks.
- §O LLM Usage Statement.

## A    EXPERIMENTAL DETAILS

We employ LoRA (Hu et al., 2022) for instruction tuning, injecting low-rank adapters with rank $r = 8$, scaling factor $\alpha = 32$, and 10% dropout. All bias parameters are kept frozen during training. For optimization, we use the AdamW optimizer (Loshchilov & Hutter, 2017) at a peak learning rate of $2 \times 10^{-5}$ and a weight decay of 0.1. The schedule begins with a linear warm-up from $1 \times 10^{-7}$ over the first 1000 steps, after which the learning rate decays exponentially to zero. Global gradient clipping with a threshold of 1.0 is employed to maintain training stability. For validation on JanusPro (Chen et al., 2025b), which requires $384 \times 384$ input images, we configure the input with $M = 1$, height $H = 16$, width $W = 9216$, and $C = 3$ channels. This setup corresponds to a square image of resolution $384 \times 384$, ensuring compatibility with the vision encoder.

## B    DOWNSTREAM TASK EVALUATION DETAILS

For language understanding, we evaluate on MMLU (Hendrycks et al., 2021) using a zero-shot setup, and on SST5 (Socher et al., 2013) with 5-shot sampling. For question answering tasks (TriviaQA (Joshi et al., 2017), NQ (Kwiatkowski et al., 2019), and PopQA (Mallen et al., 2023)), we employ Contriever (Izacard et al., 2022) to retrieve the top-$k$ relevant passages from Wikipedia, following the CEPE protocol (Yen et al., 2024). We prioritize providing the most relevant passages to the decoder to improve performance. All latency measurements are reported on a V100 GPU. **Multilingual Dataset Details.** For high-resource languages, we finetune SEETOK on ALMA (Xu et al., 2024). ALMA collects human-written test datasets from WMT'17 to WMT'20, plus the development and test sets from Flores-200 (), resulting in a total of 58K training examples across all languages. For low-resource languages, following X-ALMA (Xu et al., 2025), we use the Flores-200 dev set (Costa-Jussà et al., 2022)as our training data to ensure the quality.

**Multilingual Dataset Details.** For high-resource languages, we finetune SEETOK on ALMA (Xu et al., 2024), which collects human-written test datasets from WMT'17 to WMT'20, plus the development and test sets from Flores-200 (costa2022no), resulting in a total of 58K training examples across all languages. For low-resource languages, in line with X-ALMA (Xu et al., 2025), we use the Flores-200 development set (Costa-Jussà et al., 2022) as training data to maintain high data quality.

## C  PROMISING DIRECTION: THE VISION-CENTRIC PARADIGM

The primary computational bottleneck in current LLMs arises from their **large parameter sizes** and the **quadratic complexity** of Transformer self-attention with respect to **input length**. Text detokenization at the output stage has relatively little effect on overall model efficiency. We therefore focus on **reducing input token length** via our vision-centric tokenization while retaining the pre-trained text detokenization. This strategy not only provides substantial efficiency improvements but also enables **effective reuse of pretrained LLM knowledge and avoids catastrophic forgetting**. Importantly, it **preserves the existing MLLM architecture**, thereby allowing seamless application of our method to more vision-encoder-based MLLMs. Furthermore, the computational cost of applying our method—fine-tuning MLLMs with LoRA—is negligible compared to training a large language model from scratch for visual-text processing, which offers a **straightforward and efficient way to convert conventional text tokenization into a vision-centric tokenization scheme**. Besides, reusing the standard text detokenizer offers clear benefits. ❶ Reusing text detokenizer allows the model to retain the strong generation capabilities of the base LLM and avoids catastrophic forgetting. ❷ The use of text detokenizer eliminates the need for external OCR systems, thereby avoiding additional complexity and potential sources of error. ❸ Reusing text detokenizer preserves the existing MLLM architecture, thereby allowing seamless application of our method to more vision-encoder-based MLLMs.

A compelling but underexplored avenue is to move beyond text-tokenizer-based language models and adopt a **fully vision-centric paradigm**, where a single visual model can simultaneously process multiple modalities, such as images, text, and audio. CLIPPO (Tschannen et al., 2023) exemplifies this concept by using a single vision transformer to process images and text jointly, achieving performance on par with CLIP-style models (Radford et al., 2021) while **halving the parameter count**. Nonetheless, making a fully vision-centric paradigm effective on more complex tasks remains an open challenge, requiring further exploration in training strategies and architectural design to unlock its full potential. One possible approach is to couple the LLM with a diffusion-based decoder (as in BLIP3-o (Chen et al., 2025a)), where the LLM first produces intermediate visual features, and the diffusion model decodes them into text-rich or natural images. This would eliminate reliance on a discrete text vocabulary. However, building such a unified vision-based output module is a significant research challenge and lies beyond the scope of the current paper.

## D  EFFECTIVENESS UNDER LIMITED DATA

To assess the robustness of SEETOK under limited data, we conduct experiments on the **small** instruction-tuning dataset ALPAGASUS (Chen et al., 2024), which contains only 9k instruction–answer pairs. As shown in Table 8, our method with Qwen2.5-VL 3B (Bai et al., 2025) **achieves substantial improvements even in a low-data regime**, *e.g.*, a **+8.91** gain on the MMLU dataset (Hendrycks et al., 2021). Furthermore, scaling to larger instruction-tuning datasets leads to even more pronounced gains (*e.g.*, **+20.21** on MMLU in Table 1), demonstrating that SEETOK is effective both in low-data scenarios and when more data are available.

Table 8: Evaluation of SEETOK with Qwen2.5-VL 3B on only 9k instruction–answer pairs from ALPAGASUS, showing that our method significantly enhances visual-text instruction-following.

| Models | Text Source | TriviaQA | MMLU |
| --- | --- | --- | --- |
| Qwen2.5-VL 3B | Visual-Text | 37.55 | 32.31 |
| SEETOK | Visual-Text | 40.27(**2.72↑**) | 41.22(**8.91↑**) |

## E  SCALING STUDY ON VISUAL-TEXT INSTRUCTION DATA

We conduct a controlled scaling study by training SEETOK with 9k, 145k, and 658k visual-text instruction examples sampled from OpenHermes 2.5 (Teknium, 2023). As shown in Table 9, performance improves consistently and monotonically across all benchmarks as the amount of visual-text data increases.

Table 9: Performance improvements from scaling visual-text instruction data.

| Training Size | TriviaQA | NQ | MMLU | SST-5 |
|---|---|---|---|---|
| 0k | 37.55 | 21.13 | 32.31 | 25.21 |
| 9k | 40.27 | 22.31 | 41.22 | 30.60 |
| 145k | 42.18 | 23.16 | 49.00 | 32.80 |
| 658k | **43.53** | **24.14** | **52.52** | **44.40** |

Table 10: Comparison of FLOPs and memory usage between SEETOK and the text-tokenized QwenVL 2.5 3B.

| Model | Text Token Num | FLOPs | Memory |
|---|---|---|---|
| SEETOK | 50k | 129.55 TFLOPs | 15.6 GB |
| SEETOK | 74k | 183.97 TFLOPs | 23.6 GB |
| QwenVL 2.5 3B | 50k | 308.59 TFLOPs | 23.6 GB |

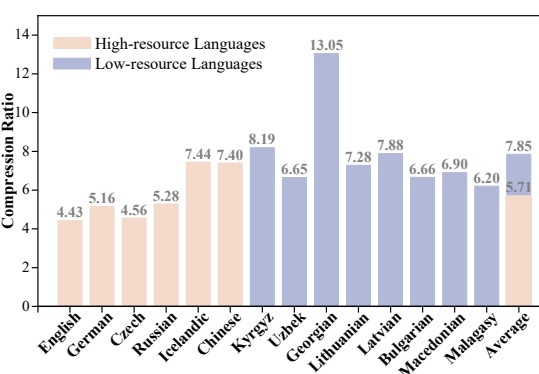

Figure 7: Compared with the text tokenization, our vision-centric tokenization achieves a compression ratio of **5.71**× in high-resource languages and **7.85**× in low-resource languages, significantly reducing sequence length.

## F  MEMORY AND COMPUTATIONAL COSTS

To quantify the efficiency advantages of SEETOK relative to the text-tokenized baseline, we analyze FLOPs and memory usage under varying input lengths. In Table 10, SEETOK demonstrates substantially lower FLOPs and memory consumption than its text-tokenized counterpart. Importantly, under a fixed memory budget of 23.6 GB, SEETOK can process up to 74k tokens, compared to 50k tokens for the QwenVL 2.5 3B baseline. This highlights the ability of SEETOK to handle significantly longer contexts within the same compute and memory constraints.

## G  COMPRESSION RATIO ACROSS HIGH- AND LOW LANGUAGES

Figure 7 presents the compression ratio across languages. We calculate this ratio on the Flores-200 test set (Costa-Jussà et al., 2022) as the average length of sequences tokenized by standard text tokenization (Qwen2.5-VL 3B (Bai et al., 2023)) divided by the average length of the same sequences tokenized by our vision-centric tokenization (SEETOK).

## H  VISUALIZATION OF VISUAL-TEXT AND LOW-RESOURCE LANGUAGES

**Visual-Text.** We visualize Visual-Text by rendering the following text as images with font sizes of 7 and 10: "*Aoccdrnig to a rscheearch at Cmabrigde Uinervtisy, it deosn't mttaer in waht oredr the*

Figure 8: Examples of Visual-Text. The text is rendered as images at font sizes 7 and 10, and all images are resized to $224 \times 224$ pixels.

**ky (Kyrgyz)**

"ky": Уч сезон мурун 28 жаштагы Видаль Севильядан Барчага кошулган.
"en": 28-year-old Vidal had joined Barça three seasons ago, from Sevilla."

**uz (Uzbek)**

"uz": Uchuvchi bo'linma rahbari Dilokrit Pattava ekanligi aniqlandi.
"en": The pilot was identified as Squadron Leader Dilokrit Pattavee.

**ka (Georgian)**

"ka": სამი სეზონის წინ ბარსას შეუერთდა 28 წლის ვიდალი სევილიიდან.
"en": 28-year-old Vidal had joined Barça three seasons ago, from Sevilla.

**lt (Lithuanian)**

"bg": "Видал е играл 49 мача за клуба, откакто се премести в Каталунската столица.",
"en": "Since moving to the Catalan-capital, Vidal had played 49 games for the club."

**lv (Latvian)**

"lv": Šajā grūtajā laikā mēs domās esam kopā ar Frenka draugiem un ģimeni
"en": Our thoughts and condolences are with Frank's family and friends at this difficult time.

**bg (Bulgarian)**

"bg": Той наскоро загуби от Раоник на открития турнир в Бризбейн
"en": He recently lost against Raonic in the Brisbane Open.

**mk (Macedonian)**

"mk": Утврдено е дека пилот бил Дилокрит Патави, водачот на ескадрилата.
"en": The pilot was identified as Squadron Leader Dilokrit Pattavee.

**mg (Malagasy)**

"mg": Vao resin'i Raonic vao haingana izy tamin'ny Brisbane Open
"en": He recently lost against Raonic in the Brisbane Open.

Figure 9: Visualization of text samples from low-resource languages

*ltteers in a wrod are, the olny iprmoetnt tihng is taht the frist and lsat ltteer be at the rghit pclae.*".
All images are resized to $224 \times 224$ pixels, shown in the Figure 8.

**Low-resource Languages.** Figure 9 illustrates examples from several low-resource languages, including Kyrgyz (ky), Uzbek (uz), Georgian (ka), Lithuanian (lt), Latvian (lv), Bulgarian (bg), Macedonian (mk), and Malagasy (mg), together with their English translations.

# I   PERTURBATION IMPLEMENTATION DETAILS

In this paper, we consider three types of perturbations: character-level, visual attacks, and word-level. **Character-level Perturbation.** Following Chai et al. (2024a), we shuffle characters within word boundaries using $n$-grams of sizes 2, 3, and 5 with a probability of 50%. We also apply $n$-gram noise by randomly inserting, deleting, or replacing characters, spaces, and punctuation marks to simulate spelling noise. This corruption occurs with a probability of 30%. Examples are shown in the Table 14.

**Word-level Perturbation.** To assess model robustness, words are randomly perturbed with probabilities $p \in \{0.2, 0.4, 0.6\}$ through synonym substitution, internal word reordering, and deletions. Examples are shown in the Table 14. Across both **word- and character-level perturbations**, the *similarity scores obtained using vision-centric tokenization consistently outperform those from text tokenization*, demonstrating its superior robustness. More examples of word-level perturbations in Chinese are provided in Figure 11.

**Visual Attack.** In line with Eger et al. (2019), each of the 26 uppercase and lowercase letters is substituted with a visually similar letter at varying probabilities $p \in \{0.2, 0.4, 0.6\}$, shown in Table 11.

Table 11: Visualization of visual attack.

| Input | Visual Attack |
|-------|---------------|
| a | â |
| b | ḃ |
| c | ĉ |
| H | Ĥ |

# J   ADDITIONAL ANALYSIS ON FAIRNESS OF PERTURBATION EVALUATION

We performed additional tests specifically designed to probe Stroop-style interference and font-style inconsistencies. SEETOK remains robust to both types of attacks.

**Stroop-Style Interference Evaluation.** We constructed a synthetic dataset consisting of 100 congruent samples (*e.g.*, "red" printed in red) and 100 incongruent samples (*e.g.*, "blue" printed in red). For each image, SEETOK is asked to identify the word and also report the color in which it is printed. The results are shown below. SEETOK reads the text perfectly in both settings (100% accuracy), but its color prediction drops to 86% on the incongruent set. This drop may be caused by semantic interference from the word itself and its inherent limitations in fine-grained color discrimination.

Table 12: Performance of SEETOK under congruent and incongruent Stroop-style conditions. Text recognition remains perfect across both settings, while color recognition shows moderate degradation under incongruent conditions.

| Condition | Task | Accuracy |
|-----------|------|----------|
| Congruent | Text recognition | 100% |
| Congruent | Font color recognition | 100% |
| Incongruent | Text recognition | 100% |
| Incongruent | Font color recognition | 86% |

**Font-style Mismatches.** SEETOK is finetuned with the Google Noto Sans font. At inference time, we render the same textual content using two additional, unseen font families (Arial, Georgia) and evaluate performance under the same protocol. The performance remains comparable or even slightly improves, showing that the model is not overly sensitive to changes in font style.

Table 13: Robustness of SEETOK to mismatched font styles. Although trained on Noto Sans, SEETOK maintains stable performance on unseen fonts.

| Font Type | TriviaQA | MMLU |
|-----------|----------|------|
| Noto Sans | 43.53 | 52.52 |
| Arial | 43.47 | 52.87 |
| Georgia | 43.62 | 52.36 |

## K  TYPOGLYCEMIA

Typoglycemia is a phenomenon in reading where humans can still recognize and comprehend words even when the internal letters of a word are scrambled, as long as the first and last letters remain in their correct positions. We evaluated the similarity between the scrambled and original text under vision-centric and text-based tokenization. **Our method consistently produces closer matches** to the original text, highlighting its resilience to letter-level noise, as seen in Table 14.

Table 14: Examples of text corruption with character- and word-level noise. We calculate the similarity scores between the original text and the corrupted text computed by text tokenization and vision tokenization. **Typoglycemia** refers to the phenomenon where words remain readable even when their **internal letters are scrambled**, as long as the first and last letters stay in place. Red indicates letters whose order has been changed, blue indicates letters that have been added or deleted, and green indicates letters that have been replaced with another character.

| Original Text | Corruption Type | Corrupted Text | Similarity (Text / Vision) |
|---------------|-----------------|----------------|----------------------------|
| The morning sun filtered through the trees, casting golden patterns on the ground. | Character-level | Teh mornnig sun fltierd trough the teers, sating godlen pattrens on the gr0nud. | 0.53 / **0.90** |
| She sipped her coffee slowly, savoring the rich aroma and warmth. | Character-level | She siped her cof fee sloowly, sav0ring the r!ch aroam and warmth. | 0.68 / **0.92** |
| The morning sun filtered through the trees, casting golden patterns on the ground. | Word-level | The ~~morning~~ sunlight filter through the ~~trees, casting golden patterns~~ the on green. | 0.69 /**0.81** |
| She sipped her coffee slowly, savoring the rich aroma and warmth. | Word-level | She ~~sipped~~ her java slowly, the rich people warmth and. | 0.61 / **0.86** |
| Human mind does not read every letter by itself, but the word as a whole. | Typoglycemia | Huamn mnid deos not raed ervey lteter by istlef, but the wrod as a wlohe. | 0.60 / **0.88** |
| According to a research team at Cambridge University, it doesn't matter in what order the letters in a word are, the only important thing is that the first and last letter be in the right place. | Typoglycemia | Aoccdrnig to a rscheearch at Cmabrigde Uinervtisy, it deosn't mttaer in waht oredr the ltteers in a wrod are, the olny iprmoetnt tihng is taht the frist and lsat ltteer be at the rghit pclae. | 0.71 / **0.88** |

## L  COMPOSITIONALITY ACROSS LANGUAGES

Visualization of subword compositionality across multiple languages ( English and Russian) is presented in Figure 10. The results show that vision-centric tokenization, compared to standard subword text tokenization, more effectively captures the hierarchical structure of language and demonstrates stronger compositional capabilities.

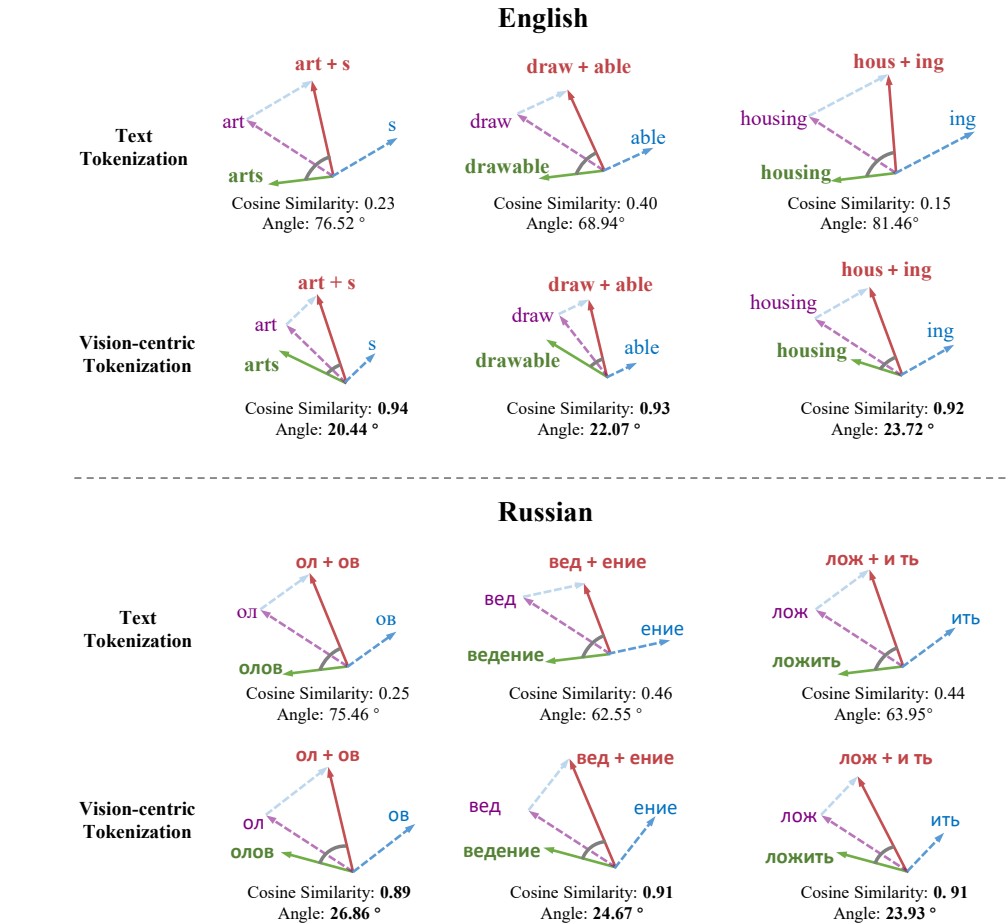

Figure 10: Visualization of subword compositionality across English and Russian. Vision-centric tokenization captures hierarchical relationships and compositional structure more effectively than subword tokenization.

| Original Text | Corrupted Text |
|---|---|
| 他静静地走在林间的小路上，月光透过稀疏的树叶洒在地面上，仿佛在为他的思绪铺开一条银色的河流。 | 他安静地在走林间路的上小，光月透过叶的稀疏树洒在地面上，在她佛仿为的绪思开一条银色铺的河流。 |
| 劝君更尽一杯酒，西出阳关无故人 | 君劝需尽一杯酒，阳关西出无故人 |

Figure 11: Visualization of word-level perturbations in Chinese sentences (including synonym substitution, deletion, and reordering). Red indicates letters whose order has been changed, blue indicates letters that have been added or deleted, and green indicates letters that have been replaced with another character.

## M ABLATION ON FINE-TUNING SCOPE

We study the effect of applying LoRA adapters to different components of the MLLM, including the vision encoder, projector, and LLM. As shown in Table 15, the best performance is achieved when adapting the vision encoder and the LLM while freezing the projector. Tuning the vision encoder alone or tuning both the vision encoder and the LLM lead to clear performance improvements In contrast, updating the projector together with other modules leads to noticeable degradation. This is because the projector has already been pretrained on large-scale image–text corpora, yielding a well-aligned interface between visual and textual features. Fine-tuning it only on the comparatively narrow instruction-tuning data tends to disrupt this alignment, resulting in worse overall performance.

Table 15: Ablation on fine-tuning scope. Keeping the projector frozen is critical for stable gains, with tuning the vision encoder and LLM providing optimal performance.

| Vision Encoder | Projector | LLM | TriviaQA |
|:---:|:---:|:---:|:---:|
| | | | 37.55 |
| ✓ | | | 40.12 |
| | ✓ | | 31.93 |
| | | ✓ | 34.16 |
| | ✓ | ✓ | 32.92 |
| ✓ | ✓ | ✓ | 37.02 |
| ✓ | | ✓ | **43.53** |

---

**Summarization**

**Visual-text Input:** "Read the following article and generate a concise summary that captures the main points in a single sentence: Scientists have discovered a new species of frog in the Amazon rainforest. The frog has unique markings and a distinct mating call. Researchers are studying its habitat and behavior to understand its ecological role."

**Answer:** "A new frog species with unique markings and calls has been discovered in the Amazon, and researchers are studying its habitat and behavior."

---

**Character Count**

**Visual-text Input:** "How many times does the letter r appear in 'strawberry'?"

**Answer:** "The letter r appears **3** times."

**Visual-text Input:** "How many times does the letter n appear in 'communion'?"

**Answer:** "The letter n appears **2** times."

---

**Recognizing Textual Entailment**

**Visual-text Input:** "Determine the relationship between the premise and the hypothesis: entailment, contradiction, or neutral. Premise: "A man is playing the guitar on stage." Hypothesis: "A person is performing music in front of an audience." "

**Answer:** "The relationship is Entailment."

---

Figure 12: Performance of SEETOK across Summarization, Character Count, and Recognizing Textual Entailment tasks. The results highlight the ability of our method to generalize effectively to diverse text processing challenges.

# N    GENERALIZATION OF SEETOK TO ADDITIONAL TASKS

We evaluate SEETOK with Qwen2.5-VL 3B across a broader set of tasks, including Summarization, Character Count, and Recognizing Textual Entailment, as illustrated in Figure 12. The summarization task demands semantic abstraction and sentence-level understanding, the Character Count task requires precise recognition of each character, and Recognizing Textual Entailment involves logical reasoning over text. The results demonstrate that our vision-centric tokenization generalizes effectively to diverse text-based tasks.

## O    LLM USAGE STATEMENT

We employed large language models (LLMs) as auxiliary tools during manuscript preparation. Their use was strictly limited to surface-level editing tasks, including grammar correction, minor rephrasing, and stylistic improvements to enhance readability. At no point did we rely on LLMs for generating research ideas, methods, experiments, or conclusions. All technical content and analysis presented in this paper are the sole work of the authors.

