# OpenReview forum: "See the Text: From Tokenization to Visual Reading"
_ICLR.cc/2026/Conference — Submitted to ICLR 2026_

### Official Review · Reviewer_Kw1w · 2025-10-27

**Soundness:** 3
**Presentation:** 3
**Contribution:** 2
**Rating:** 4
**Confidence:** 4

**Summary:**

This paper challenges the dominant subword tokenization paradigm in LLMs by proposing SEETOK, a vision-centric alternative inspired by human reading. Instead of breaking text into discrete subword units, SEETOK renders text as an image and leverages the vision encoder of a pretrained Multimodal Large Language Model (MLLM) to process it. This "visual tokenization" is followed by a lightweight LoRA-based instruction tuning phase to adapt the MLLM to understand instructions presented in this visual-text format.

**Strengths:**

The experimental validation is extensive and convincing. The authors evaluate on a diverse set of tasks (QA, translation, sentiment) and languages (13+), and perform deep-dive analyses into efficiency (FLOPs, latency), robustness (multiple perturbation types), and compositionality. The results are consistently strong across the board.

**Weaknesses:**

A similar idea in VisInContext: Leveraging Visual Tokens for Extended Text Contexts in Multi-Modal Learning.

**Questions:**

Please see the weaknesses.

---

> ### Author Response · Authors · 2025-11-22
> **Point-to-Point Response to Reviewer Kw1w**
>
> We're glad you found the experimental validation convincing and the strong performance of SEETOK! We provide point-to-point response below.
>
> **Q1: Similar idea in VisInContext.**
>
> **A1:** We respectfully disagree. Although both works use visual representations of text, they address fundamentally different research problems.  We outline the distinctions in terms of motivation, research topic, and technical design.
> 1. **Motivation**
> - VisInContext addresses computational bottlenecks in multimodal long-context scenarios by using visual tokens to extend the effective context length. SEETOK, by contrast, tackles fundamental issues of text tokenization (e.g., vocabulary constraints and perturbation sensitivity) and explores a vision-centric alternative to overcome them.
>
> 2. **Research topic**
> - VisInContext targets multimodal in-context learning tasks, while SEETOK is developed for pure-text language tasks.
>
> 3. **Technical design**
> - SEETOK converts all text into visual inputs and thus removes the need for text tokenization. VisInContext, however, still requires text tokenization and mixes three input types: natural images, pure text, and rendered text images.
> - VisInContext relies on text-tokenized embeddings to help the vision encoder interpret the visualized text. SEETOK instead uses vision-centric instruction tuning alone, without requiring text-tokenized embeddings, and substantially improves visual-text instruction following.
> - VisInContext builds upon the Flamingo model, where visual features from the vision encoder are fed into the LLM through cross-attention layers. SEETOK, however, is built upon modern multimodal MLLMs (e.g., QwenVL2.5), where the visual features are passed directly to the LLM.
>
> ---
>
> Thank you for your time and feedback. We hope we addressed your concerns. Please let us know if you'd like any further information!

---

### Official Review · Reviewer_DoBY · 2025-10-28

**Soundness:** 3
**Presentation:** 3
**Contribution:** 2
**Rating:** 4
**Confidence:** 4

**Summary:**

This paper introduces SEETOK, a vision-centric tokenization method that renders textual input as images and leverages pretrained vision-language models (MLLMs) to process language tasks through visual pathways. The authors propose treating the vision encoder and projector as a pluggable “visual tokenizer,” which replaces traditional subword tokenization. Coupled with lightweight LoRA-based instruction tuning, SEETOK enables existing MLLMs to process textual tasks with reduced token count, lower FLOPs, and improved robustness to orthographic perturbations. The paper presents comprehensive evaluations covering multilingual efficiency, perturbation robustness, and compositionality, and conducts hierarchical geometric analysis via orthogonal Procrustes to support the claims.

**Strengths:**

1. Clear engineering contribution: The paper proposes a practical and well-motivated framework that treats rendered text as input for vision encoders in existing MLLMs. This vision-as-tokenizer approach is modular, minimally invasive, and compatible with off-the-shelf MLLMs, requiring only LoRA-based tuning without retraining the full model.

2. System-level benefits: The authors go beyond token count reduction to quantify end-to-end FLOPs and latency improvements (e.g., 70.5% FLOP reduction and 33.5% latency drop on TriviaQA), making a compelling case for the method’s efficiency in real-world deployments.

3. Solid empirical support: The experimental section is thorough, with multilingual translation, robustness testing, compositionality probing, and ablation studies. The reported results indicate competitive performance in multilingual and perturbed settings, as well as encouraging signs of compositional alignment.

**Weaknesses:**

**Major Issues**

1. Incremental novelty: While the integration into MLLMs is elegant, the central idea, rendering text as images and processing it via vision encoders, has been extensively explored in prior work (e.g., PIXEL, CLIPPO, and CLIP-style MLLMs). Many of the claimed benefits (e.g., robustness to perturbation, fertility reduction, multilingual fairness) are inherent not specific to SEETOK. The primary contribution lies in integrating this paradigm into general-purpose MLLMs with lora finetuning.

2. Lack of evaluation on vision-native tasks: The paper does not assess whether SEETOK compromises the model’s original visual reasoning capabilities, particularly on standard benchmarks such as VQA. This raises questions about whether the instruction tuning deteriorates native visual-language performance.

3. Underwhelming absolute accuracy: On some core benchmarks like MMLU (Table 5 and 6), SEETOK lags behind pure-text tokenization, with the performance gap only partially compensated by its efficiency or robustness advantages. The narrative leans heavily on efficiency and multilinguality, but the trade-off in task-specific accuracy should be more candidly discussed.

4. Missing citations: Key related works are absent, such as:

[1] Textural or Textual: How Vision-Language Models Read Text in Images, ICML 2025.

[2] How Do Large Vision-Language Models See Text in Image? Unveiling the Distinctive Role of OCR Heads, EMNLP 2025.

**Minor Issues**

1. Fairness of perturbation evaluation: In Section 4.4, the comparison between pure-text and vision-centric models on homoglyph or character attacks may be inherently biased. Pure-text models are expected to be more sensitive to glyph-level noise, whereas vision-based models may be more vulnerable to stroop-style interference (e.g., visually red “blue”) or font-style mismatches. These asymmetric vulnerabilities deserve more nuanced analysis.

2. Missing evaluation on LLaVA-family models: The current experiments primarily cover Qwen2.5 and JanusPro backbones. Adding results on LLaVA or other open-source MLLMs could further validate generalizability.

**Questions:**

1. Capability retention on standard VQA tasks: How does SEETOK impact performance on vision-native tasks (e.g., VQA, TextVQA, DocVQA)? Including such evaluations would strengthen the claim of SEETOK as a general-purpose plug-in.

2. Finer-grained ablation: Table 4 reports the effect of tuning different components. Could the authors include intermediate ablations, e.g., “only tune projector” vs. “only tune vision encoder”? This would clarify the contribution of each module.

3. Absolute accuracy plots for perturbation: Figure 4 shows accuracy drops, but absolute accuracy values under each noise setting would help readers gauge practical utility more clearly.

4. Compositional proximity analysis: Similar to Figure 5 (left), it would be interesting to report whether SEETOK embeddings of "lemon" are closer to lime or demon.

---

> ### Author Response · Authors · 2025-11-22
> **Point-to-Point Response to Reviewer DoBY (1/3)**
>
> We are grateful that the reviewer recognized our practical and well-motivated design and found the experimental section thorough. Thank you for your constructive feedback. We provide point-to-point response below.
>
> **Q1: Incremental novelty.**
>
> **A1:** We respectfully disagree. We would like to clarify that SEETOK introduces a new paradigm for vision-centric text processing—repurposing the vision encoder of MLLM as a promising alternative to text tokenization, fundamentally different from prior work. This design enables SEETOK to match and even surpass text-tokenized baseline, while simultaneously delivering substantial efficiency gains. Taken together, these results position SEETOK as a meaningful advancement in vision-centric text processing. Below, we clarify the key distinctions between SEETOK and existing approaches in terms of model architecture, training objective, model performance, and unique benefits.
>
> 1. **Model architecture.** PIXEL[1] is a a ViT-MAE model, which is an encoder-only architecture and cannot perform generative tasks such as open-ended text generation or sequence continuation. CLIPPO[2], a variant of CLIP, utilizes one single vision encoder to process both images and text that is rendered as images. SEETOK, by contrast, builds on MLLM that integrates a vision encoder with a decoder-only language model. To the best of our knowledge, it is the first attempt to repurpose the vision encoder of MLLM as a practical alternative to text tokenization, opening a new pathway for vision-centric text processing.
> 2. **Training objective.** PIXEL[1] is trained with masked patch prediction. CLIPPO [2] relies on CLIP-style contrastive learning. SEETOK instead introduces vision-centric instruction tuning with next token prediction, achieving significant performance improvements (e.g., +20.21 on MMLU in Table 1).
> 3. **Model performance.**  PIXEL[1] and CLIPPO[2] are encoder-only models and thus cannot support free-form generative language modeling. Accordingly, when applied to text-only tasks, they are evaluated primarily on relatively simple benchmark (e.g., GLUE) and require task-specific finetuning, yet still underperform text-only baseline such as BERT. SEETOK instead is evaluated on more challenging language benchmarks (e.g., MMLU and open-domain QA). Despite the increased task difficulty, SEETOK is competitive with, and in some cases exceeds, the text-tokenized baseline (e.g., 43.53 vs. 41.92 on TriviaQA), while also offering substantial efficiency gains (4.43× token reduction), improved cross-lingual transferability, and stronger robustness to typographic noise.
> 4. **Unique benefits.** Fertility reduction is the unique benefit of SEETOK. Importantly, vision-based text processing does not inherently guarantee shorter token sequences. For example, CLIPPO[2], as shown in Fig. 9 of [2], can `require even more tokens` than T5 tokenization (vocabulary size 32k) for high-resource languages such as English and German. SEETOK, by contrast, achieves a `4.43× reduction` in English and consistent efficiency across many other languages, even when compared against Qwen tokenization’s significantly larger vocabulary (151k). Moreover, our experiments further provide rigorous empirical evidence supporting its stronger cross-lanugage transferability in Figure 3 and Table 3. While CLIPPO[1] and PIXEL[2] report robustness advantages, they are encoder-only architectures. SEETOK differs fundamentally and we additionally show that it maintains strong robustness within a generative MLLM.
>
> In summary, we believe our work offers valuable new insights, technical contributions, and meaningful experiments to the field of processing text visually.
>
> [1] Language Modelling with Pixels. ICLR 2023.
>
> [2] CLIPPO: Image-and-Language Understanding from Pixels Only. CVPR 2023.
>
> **Q2: Lack of evaluation on vision-native tasks.**
>
> **A2:** Great suggestion! We test SEETOK on VQAv2, TextVQA, and DocVQA. As shown below, its performance remains on par with Qwen2.5-VL 3B, confirming that our instruction tuning does not harm the native vision–language performance. We add these results in Table 7.
> |Model|VQAv2|DocVQA|TextVQA
> |:-:|:-:|:-:|:-:|
> |Qwen2.5-VL 3B|81.2|93.9|79.3
> |SEETOK|81.0|93.5|80.1

---

> > ### Author Response · Authors · 2025-11-22
> > **Point-to-Point Response to Reviewer DoBY (2/3)**
> >
> > **Q3: Underwhelming absolute accuracy.**
> >
> > **A3:** Thank you for raising this point. We would like to clarify that we had already discussed the performance gap on MMLU in Lines 261–265 of the original paper. As discussed, SEETOK has far less exposure to knowledge-dense examples in visual-text form compared to large-scale text-only pretraining, which leads to the performance gap. Encouragingly, this `gap narrows as visual-text training scales`: increasing training data from 9k to 145k and 658k yields steady improvements across all benchmarks, indicating significant potential to further close or even exceed this gap. We include the analysis in Sec. 4.2 and add the full scaling results in Appendix E.
> >
> > |Training Size|TriviaQA|NQ|MMLU|SST5
> > |:-:|:-:|:-:|:-:|:-:|
> > |-|37.55|21.13|32.31|25.21
> > |9k|40.27|22.31|41.22|30.60
> > |145k|42.18|23.16|49.00|32.80
> > |658k|43.53|24.14|52.52|44.40
> >
> > **Q4: Missing citations**
> >
> > **A4:** Thanks for your careful review! Textural-or-Textual [3] explores how vision-language models (e.g., CLIP) process text in images using the proposed ToT dataset, showing that early layers mainly rely on visual texture, with semantic understanding emerging only in the final blocks. Paper [4] identifies specialized “OCR heads” within MLLMs that are responsible for reading text directly from images. They provide valuable insights into how existing multimodal models internally process text in visual inputs. We incorporate citations to both works into the related work section.
> >
> > [3] Textural or Textual: How Vision-Language Models Read Text in Images, ICML 2025.
> >
> > [4] How Do Large Vision-Language Models See Text in Image? Unveiling the Distinctive Role of OCR Heads. EMNLP 2025.
> >
> > **Q5: Fairness of perturbation evaluation.**
> >
> > **A5:** Good comment! Following your suggestion, we perform additional tests specifically designed to probe Stroop-style interference and font-style inconsistencies. We find that SEETOK remains robust to both types of attacks.
> > 1. **Stroop-style interference:** We constructed a synthetic dataset consisting of 100 congruent samples (e.g., “red” printed in red) and 100 incongruent samples (e.g., “blue” printed in red). For each image, SEETOK is asked to identify the word and also report the color in which it is printed. The results are shown below. SEETOK reads the text perfectly in both settings (100% accuracy), but its color prediction drops to 86% on the incongruent set. This drop may be caused by semantic interference from the word itself and its inherent limitations in fine-grained color discrimination.
> >
> > |Condition|Task|Accuracy|
> > |:-:|:-:|:-:|
> > |Congruent|Text recognition|100%|
> > |Congruent|Font color recognition|100%|
> > |Incongruent|Text recognition|100%|
> > |Incongruent|Font color recognition|86%|
> >
> > 2. **Font-style mismatches:** SEETOK is finetuned with the Google Noto Sans font. At inference time, we render the same textual content using two additional, unseen font families (Arial, Georgia) and evaluate performance under the same protocol. The performance remains comparable or even slightly improves, showing that the model is not overly sensitive to changes in font style.
> >
> > |Font Type|TriviaQA|MMLU|
> > |:-:|:-:|:-:|
> > |Noto Sans|43.53|52.52|
> > |Arial|43.47|52.87|
> > |Georgia|43.62|52.36|
> >
> > 3. **Comprehensive perturbation evaluation:** Note that our perturbation analysis in Section 4.4 is not limited to visual or homoglyph attacks. It also includes a wide range of character-level and word-level perturbations—such as insertion, deletion, substitution, and shuffling—providing a more comprehensive robustness evaluation than focusing solely on glyph-level distortions.
> >
> > We add these discussions in Appendix J.
> >
> > **Q6: Evaluation on LLaVA-family model.**
> >
> > **A6:** Thank you for the suggestion. To further assess the generalizability of SEETOK, we additionally apply it to LLaVA-Next-8B. As shown below, SEETOK substantially improves performance on TriviaQA, demonstrating that our approach transfers well beyond Qwenvl- and Janus-based MLLMs. We add the results in Table 5.
> >
> > |Model|TriviaQA
> > |:-:|:-:
> > |Llava-next-8b|51.18
> > |SEETOK|**59.72**
> >
> > **Q7: Finer-grained ablation.**
> >
> > **A7:** Thank you for the suggestion. We include the finer-grained ablations in Appendix M. The results show that tuning the vision encoder alone or tuning both the vision encoder and the LLM leads to clear performance improvements, while tuning the projector tends to degrade performance.
> > |Vision Encoder|Projector|LLM|TriviaQA|
> > |-|-|-|-|
> > | | | |37.55|
> > |✔|||40.12|
> > ||✔||31.93|
> > |||✔|34.16|
> > ||✔| ✔ |32.92|
> > |✔|✔|✔|37.02|
> > |✔||✔|**43.53**|
> >
> > **Q8: Absolute accuracy plots for perturbation.**
> >
> > **A8:** Thank you for the helpful suggestion. We update Figure 4 accordingly to include absolute accuracy values for all perturbation types.

---

> ### Author Response · Authors · 2025-11-22
> **Point-to-Point Response to Reviewer DoBY (3/3)**
>
> **Q9: Compositional proximity analysis.**
>
> **A9:** Cosine similarity computed from SEETOK embeddings shows that ⟨lemon, lime⟩ = 0.64, ⟨lemon, demon⟩ = 0.57. Despite the greater visual similarity between lemon and demon, the lower similarity may be because SEETOK incorporates both visual and semantic cues in its embedding space, which is consistent with insights from Textural-or-Textual[3].
>
> [3] Textural or Textual: How Vision-Language Models Read Text in Images, ICML 2025.
>
> ---
>
> We appreciate again your thoughtful review and we hope we addressed your concerns. Please let us know if you'd like any further information!

---

### Official Review · Reviewer_QVXp · 2025-10-31

**Soundness:** 2
**Presentation:** 3
**Contribution:** 3
**Rating:** 4
**Confidence:** 3

**Summary:**

The paper proposes an approach to replace all the text tokens with vision tokens of the text image in vision language models for language-only tasks, and conducts comprehensive experiments to evaluate the advantages of doing so. Their experiments show that after further instruction-tuning, models using visual-text tokens achieve performance comparable to those using pure text tokens across various language-only benchmarks. Moreover, they show that such an approach may enable more holistic text perception, shown by less performance drop when perturbing the elements in the text.

**Strengths:**

The idea of using pure visual tokens as input is well motivated. The experiments to reveal the benefit of using visual-centric tokenization are comprehensive. The Perturbation Probing study is an interesting investigation that reveals the unique advantage of holistic perception using visual tokens to represent text.

**Weaknesses:**

1. My major concern remains whether the performance gain comes from the finetuning process or SEETOK itself,  especially considering the low performance on SST5 for the original Qwen2.5-VL 3B model. Although it’s discussed in Table 6, it would strengthen the paper’s conclusion to also show the evaluation results for all the datasets evaluated in Table 1 and Table 3, rather than only showing the results of MMLU. I would be more curious about the results of TriviaQA and SST5.

2. Line 429: “even though the finetuning was performed using visual-text data, the model benefits from better cross-format generalization, enhancing its pure text performance. ”. It’s also hard to get such a conclusion, given that finetuning with the same data already has +0.30 improvement, and it’s even under a reduced training dataset size. Would expect to see similar results evaluated by other datasets to make the conclusion solid.



Minor: Page 9 has too much bold text that actually influences the reading. I don’t think method names like “Orthogonal Procrustes Analysis” and “residual norm” really need to be bolded.

**Questions:**

Question:
For Table 6, what does it mean to have SEETOK with pure-text as both training input and inference input (line 4)? I thought the whole point of SEETOK is to utilize the visual token as a representation of text and finetune both the vision encoder and LLM decoder. This should be just considered as a LORA fine-tuning for the baseline model, or does it have anything different from a common fine-tuning process?

---

> ### Author Response · Authors · 2025-11-22
> **Point-to-Point Response to Reviewer QVXp**
>
> Glad to see the motivation resonated with you! Thank you for your insightful review. We provide point-to-point response below.
>
> **Q1: Whether the performance gain comes from the finetuning process or SEETOK itself.**
>
> **A1:** Great comment! To strengthen the conclusion that the performance gain primarily comes from SEETOK rather than the finetuning process, we finetune Qwen2.5-VL 3B on the same OpenHermes 2.5 dataset in both visual–text and pure-text settings, and evaluate both models on all datasets in Tables 1 and 3. The visual–text setup `consistently achieves larger gains`, indicating that the improvements are attributable to SEETOK rather than finetuning alone. We add the full results to Table 3 and Table 4 in the revised paper. (Table 4 in the revised paper corresponds to Table 6 in the original paper.)
>
> - Table 4
>
> |Models|Training Input|Inference Input|TriviaQA|NQ|PopQA|SST5|
> |:-:|:-:|:-:|:-:|:-:|:-:|:-:|
> |Qwen2.5-VL 3B|-|Pure-Text|41.92|29.31|24.64|28.80
> |Qwen2.5-VL 3B$^{*\clubsuit}$|Pure-Text|Pure-Text|42.06(+0.14)|29.75(+0.44)|24.96(+0.32)|30.00(+1.20)
> |Qwen2.5-VL 3B|-|Visual-Text|37.55|21.13|20.16|25.21
> |+SEETOK*|Visual-Text|Visual-Text|42.18(**+4.63**)|23.16(**+2.03**)|23.47(**+3.31**)|32.80(**+7.59**)
>
> - Table 3
>
> |Models|Training Input|Inference Input|de|cs|is|zh|ru
> |:---:|:---:|:---:|:---:|:---:|:---:|:---:|:---:|
> |Qwen2.5-VL 3B|-|Pure-Text|67.25|62.02|53.63|57.51|63.16
> |Qwen2.5-VL 3B$^{*\clubsuit}$|Pure-Text|Pure-Text|67.88(+0.63)|62.05(+0.03)|53.89(+0.26)|58.12(+0.61)|65.33(+2.17)
> |Qwen2.5-VL 3B|-|Visual-Text|47.49|41.02|34.37|46.77|46.44
> |+SEETOK*|Visual-Text|Visual-Text|65.63(**+18.14**)|64.89(**+23.87**)|54.97(**+20.60**)|68.94(**+22.17**)|71.42(**+24.98**)
>
> ∗ indicates results on a reduced training dataset, where long samples were removed to prevent out-of-memory issues with pure-text input. Qwen2.5-VL 3B$^{\clubsuit}$ refers to the pure-text finetuning variant and +SEETOK* represents Qwen2.5-VL 3B finetuned under the visual–text configuration.
>
> **Q2: Similar results evaluated by other datasets to make the conclusion solid.**
>
> **A2:** Insightful suggestion! To more rigorously validate the effect of visual–text finetuning on pure text performance, we expand our evaluation to more benchmarks. As shown in the table below, with identical training data, finetuning in text-only format produces only moderate gains, whereas finetuning on the `visual–text` representation yields `larger improvements` in pure text tasks. This further supports our conclusion that visual–text finetuning provides strong cross-format generalization benefits. Notably, finetuning with visual-text inputs is more efficient, as it uses far fewer input tokens than pure-text finetuning, allowing the model to achieve stronger improvements at lower computational cost. We add the full results in Table 4.
>
> |Models|Training Input|Inference Input|TriviaQA|NQ|PopQA|SST5
> |:-:|:-:|:-:|:-:|:-:|:-:|:-:|
> |Qwen2.5-VL 3B|-|Pure-Text|41.92|29.31|24.64|28.80
> |Qwen2.5-VL 3B$^{*\clubsuit}$|Pure-Text|Pure-Text|42.06(+0.14)|29.75(+0.44)|24.96(+0.32)|30.00(+1.20)
> |+SEETOK*|Visual-Text|Pure-Text|42.54(**+0.62**)|30.18(**+0.87**)|25.21(**+0.57**)|31.42(**+2.62**)
>
> Qwen2.5-VL 3B$^{\clubsuit}$ refers to the pure-text finetuning variant and +SEETOK* represents Qwen2.5-VL 3B finetuned under the visual–text configuration.
>
> **Q3: Too much bold text.**
>
> **A3:** Thank you for the suggestion. The excessive bold formatting is removed, and method names such as Orthogonal Procrustes Analysis and residual norm are now presented in regular text to improve readability.
>
>
> **Q4: What does it mean to have SEETOK with pure-text as both training input and inference input (line 4)?**
>
> **A4:** Sorry for the confusion! In Table 4 of Row 4 (i.e., Table 6 in the original paper), the model is finetuned using the same procedure as SEETOK (LoRA applied to both the vision encoder and decoder), but training with pure-text inputs. To avoid misunderstanding, we revise the model name in Table 4 (line 4) to Qwen2.5-VL 3B$^{\clubsuit}$, where ${^\clubsuit}$ denotes the same finetuning setup as SEETOK, with pure-text training input.
>
> ---
>
> We appreciate again your thoughtful review and we hope we addressed your concerns. Please let us know if you'd like any further information.

---

### Official Review · Reviewer_Gw3z · 2025-10-31

**Soundness:** 3
**Presentation:** 3
**Contribution:** 2
**Rating:** 4
**Confidence:** 5

**Summary:**

This paper introduces SEETOK, a vision-centric tokenization method for large language models that renders text as images and leverages pretrained multimodal LLM vision encoders to interpret them. Instead of using traditional subword tokenization, SEETOK processes text visually through the vision pathway of models like Qwen2.5-VL and JanusPro. The method uses LoRA adapters for lightweight fine-tuning to enable visual-text instruction following. Experiments across natural language understanding tasks, multilingual translation, and robustness evaluations demonstrate that SEETOK achieves comparable or superior performance to text tokenization while requiring 4.43× fewer tokens, reducing FLOPs by 70.5%, and showing improved cross-lingual generalization, robustness to typographic noise, and compositional understanding.

Overall, it is an incremental paper to continue to explore the vision tokenizer for LLMs, the approach proposed in this work is okay, which tries to solve some issues in the encoder part, but the decoder part is still traditional text tokenizer.

**Strengths:**

1. The paper presents a way to leverage the visual tokenization for text processing. By leveraging the pretrained MLLMs with LoRA adaptation is efficient and pragmatic, avoiding expensive training from scratch.

2. The proposed SEETOK demonstrates good performance on several categories of tasks, including QA, translation, cross-lingual transfer etc. And also shows it robustness.

3. The proposed SEETOK tokenizer also shows its superb efficiency in token comprehension and FLOP reduction.

**Weaknesses:**

1. Overall, it is interesting to see that introducing the vision tokenization as the tokenzier for text. While, for the detokenization part, it still rely on the traditional text tokenizer, may still suffer from the existing issues for text tokenizer. This might be not a perfect point. So, how to implement a full vision-centric tokenizer here for text?

2. Table 1 shows SEETOK underperforms on MMLU (52.52 vs 61.91) and NQ (24.14 v.s. 29.31), this might be a significant limitation for knowledge-heavy tasks?

**Questions:**

There are some existing work on leveraging the vision for LM tokenization, for example, screenshot LM [1], PIXAR [2], Pix2Struct[3] etc. Maybe it is better to discuss with them, or compare with them on the shared tasks.


Another question is - how does SEETOK scale to very long sequence/documents (for example, 100K+ tokens in text form)? What happens to memory and computational costs?


1. Improving Language Understanding from Screenshots, https://arxiv.org/abs/2402.14073
2. PIXAR: Auto-Regressive Language Modeling in Pixel Space, https://arxiv.org/abs/2401.03321
3. Pix2Struct: Screenshot Parsing as Pretraining for Visual Language Understanding, https://arxiv.org/abs/2210.03347

---

> ### Author Response · Authors · 2025-11-22
> **Point-to-Point Response to Reviewer Gw3z (1/2)**
>
> We're thrilled you found our vision-centric tokenization interesting and the good performance of SEETOK. Thank you for your comprehensive review. We'll now address your valuable feedback.
>
> **Q1: Decoder part is still traditional text detokenizer.**
>
> **A1:** Insightful question! We would like to clarify that a full vision-centric tokenizer is not necessary. Input-side vision tokenization already captures the core benefits. Moreover, since the traditional text detokenizer incurs **virtually no computational overhead**, retaining it remains both an effective and practical design choice. Specifically:
> - **Vision-centric detokenization is not essential.**
>   1. The primary bottleneck in LLMs stems from large parameter counts and the quadratic attention cost over the input sequence. Hence, reducing input tokens through vision-centric tokenization yields efficiency benefits. Output-side text detokenization simply maps predicted tokens back to subword units in the vocabulary, incurring negligible computational overhead. Therefore, replacing text detokenization with a vision-centric detokenizer would `not meaningfully improve efficiency and would instead introduce unnecessary architectural complexity`.
>   2. Even with a standard text detokenizer, vision-centric tokenization still delivers key advantages such as multilingual fertility reduction and robustness to orthographic perturbations.
>
> - **Reusing the standard text detokenizer offers clear benefits.**
>   1. Reusing text detokenizer allows the model to retain the strong generation capabilities of the base LLM and avoids catastrophic forgetting.
>   2. The use of text detokenizer eliminates the need for external OCR systems, thereby avoiding additional complexity and potential sources of error.
>   3. Reusing text detokenizer preserves the existing MLLM architecture, thereby allowing seamless application of our method to more vision-encoder-based MLLMs.
>
> We had discussed the vision-centric detokenization in the original paper of Appendix C and a more detailed analysis is added accordingly.
>
> **Q2: Limitation for knowledge-heavy tasks.**
>
> **A2:** Thanks for your careful review! We had already discussed the performance gap on MMLU in Lines 261–265 of the original paper. We would like to clarify vision-centric tokenization is not inherently limited to knowledge-heavy tasks and explain the performance gap.
>
> 1. **SEETOK can outperform text-tokenized baseline on knowledge-intensive tasks.** On another critical knowledge-heavy benchmark TriviaQA, SEETOK outperforms QwenVL-2.5 3B with pure-text input (43.53 vs. 41.92) while using 4.43× fewer tokens. This demonstrates that SEETOK is a promising alternative and can achieve strong performance even in knowledge-intensive settings.
>
> 2. **SEETOK has substantial potential to close the gap on MMLU and NQ.** Knowledge-intensive benchmarks heavily rely on world knowledge acquired during massive textual pretraining. SEETOK, however, is exposed to far less such knowledge in visual-text form, causing the gap. Encouragingly, this gap narrows as visual-text training scales: increasing the data from 9k to 145k and 658k yields steady improvements across all benchmarks, suggesting clear potential to close the gap or even surpass the baseline with further scaling.
>
> |Training Size|TriviaQA|NQ|MMLU|SST5
> |:-:|:-:|:-:|:-:|:-:|
> |-|37.55|21.13|32.31|25.21
> |9k|40.27|22.31|41.22|30.60
> |145k|42.18|23.16|49.00|32.80
> |658k|43.53|24.14|52.52|44.40
>
> 3. **Balancing performance and computation.** SEETOK achieves a 4.43× token reduction compared with text tokenization, offering substantially lower computation while maintaining competitive or superior performance.
>
> We include the analysis in Sec. 4.2 and add the full scaling results in Appendix E.

---

> > ### Author Response · Authors · 2025-11-22
> > **Point-to-Point Response to Reviewer Gw3z (2/2)**
> >
> > **Q3: Discuss with some existing work.**
> >
> > **A3:** Thanks for your suggestion! We make detailed discussions with [1,2,3] separately. SEETOK is fundamentally different from [1,2,3].
> > - **Training objective.** The pretraining objectives in [1,2,3] are more complex than SEETOK, and even require multi-stage training [2,3]. Screenshot LM [1] combines image patch prediction with text token prediction; PIXAR [2] trains on next-patch prediction and adopts adversarial training; and Pix2Struct [3] learns to parse masked webpage screenshots into simplified HTML. In contrast, SEETOK relies solely on visual-text instruction tuning with next token prediction, a simple yet highly effective objective that directly improves performance on pure-text tasks.
> > - **Method framework.** Pix2Struct [3] adopts an encoder–decoder architecture. PIXAR [2] provides a compelling demonstration of a fully vision-centric design, generating text autoregressively through image patches. However, it can only produce short pixel-based text sequences and requires an additional OCR system to recover the textual output. Screenshot LM [1] explores both encoder-only and decoder-only variants, where the decoder-only setup is similar to Fuyu-style architectures that do not include image encoder. SEETOK, in contrast, builds on multimodal LLM that couples vision encoder with decoder-only LLM (e.g., QwenVL 2.5).
> > - **Model performance.** SEETOK not only performs competitively with text-tokenized baseline on challenging benchmarks (and even surpasses it on TriviaQA), but also offers efficiency improvements, lower multilingual fertility, and enhanced robustness to typographic noise. In addition, we compare SEETOK with Screenshot LM [1] and PIXAR [2] on SST2 and RTE. Although both [1] and [2] are finetuned on these datasets, SEETOK still achieves consistently better results. Pix2Struct [3] is not intended for text-only tasks and, as noted in [1], performs poorly in such settings. Therefore, following [1], we omit it from this comparison.
> >
> > |Model|SST2|RTE
> > |:-:|:-:|:-:|
> > |PIXAR|89.0|58.5
> > |screenshot LM|92.5|67.7
> > |SEETOK (Ours)|$\bf94.8$|$\bf93.2$
> >
> > We add the discussion in the related work section.
> >
> >
> > **Q4: How does SEETOK scale to very long sequence/documents.**
> >
> > **A4:** Good comment! With the same 23.6 GB memory constraint, SEETOK can handle sequences up to 74k text tokens, whereas the text-tokenized QwenVL 2.5 3B baseline can only support up to 50k tokens. Moreover, SEETOK delivers higher efficiency, requiring fewer FLOPs and lower memory usage when processing the same 50k text tokens. As processing 100k text tokens causes out-of-memory in QwenVL 2.5 3B with pure-text input, we restrict our comparison to settings that remain executable. We add the analysis to Appendix F.
> > |Model|Text Token Num|FLOPs|Memory
> > |:-:|:-:|:-:|:-:
> > |SEETOK|50k|129.55 TFLOPS|15.6 GB
> > |SEETOK|74k|183.97 TFLOPs|23.6 GB
> > |QwenVL 2.5 3B|50k|308.59 TFLOPs|23.6 GB
> >
> > [1] Improving Language Understanding from Screenshots, https://arxiv.org/abs/2402.14073
> >
> > [2] PIXAR: Auto-Regressive Language Modeling in Pixel Space, https://arxiv.org/abs/2401.03321
> >
> > [3] Pix2Struct: Screenshot Parsing as Pretraining for Visual Language Understanding, https://arxiv.org/abs/2210.03347
> >
> > ---
> >
> > We sincerely appreciate your constructive suggestions again. We hope we addressed all of your concerns. Please let us know if you require any additional information.

---

### Author Response · Authors · 2025-12-04
**Summary of Reviews and Revisions**

To all reviewers:

We express our sincere gratitude to all reviewers for their valuable time and thorough assessment of our manuscript. In response, we have carefully addressed each concern raised, and provided point-to-point clarifications which are integrated into the new version of our manuscript.

We are gratified by the positive feedback from all reviewers, especially the comments highlighting our vision-centric tokenization as interesting (Reviewer Gw3z) and well-motivated (Reviewers QVXp and DoBY), the good performance (Reviewers Gw3z and Kw1w), and the comprehensive evaluations (Reviewers QVXp, DoBY, and Kw1w).

Foremost among the concerns is the comparison with existing related methods. We clarify the key distinctions in terms of *model architecture, training objectives, model performance, and unique benefits*. To the best of our knowledge, SEETOK is the `first attempt` to repurpose the vision encoder of an MLLM as a practical alternative to text tokenization, opening a new pathway for vision-centric text processing and fundamentally differing from prior work. Previous methods that process text visually **lag significantly behind** their text-tokenized counterparts, with evaluations restricted to traditional simple pure-text task GLUE. In contrast, SEETOK is able to match or even **surpass** text-tokenized baseline on a range of widely used and challenging pure-text tasks (e.g., open-domain QA and general reasoning benchmark MMLU).

Beyond performance improvements, SEETOK also offers `unique advantages`, delivering **substantial efficiency gains, stronger cross-language transferability, and improved robustness to typographic noise**. SEETOK `generalizes effectively across multiple MLLMs`, including Qwen-VL 2.5, JanusPro, and the Llava-next. Moreover, it preserves the MLLM architecture and is able to maintain, or even enhance, native visual-language performance, providing an elegant yet highly effective drop-in solution. Taken together, these contributions position SEETOK as a **meaningful advancement** in vision-centric text processing.

Other questions, such as the use of a traditional text detokenizer, performance on knowledge-heavy tasks, scaling to very long sequence, ablation studies on more datasets, evaluation on vision-native tasks, fairness of perturbation evaluation, evaluation on LLaVA-family model, and absolute accuracy plots have also been addressed accordingly.

For more details, please refer to our responses to each reviewer. We have strived to address each of your concerns.

Sincerely yours,

Authors

---

### Meta-Review · Area_Chair_RpBS · 2026-01-06

**Summary:**

This paper proposes SEETOK, a vision-centric approach that renders text as images and uses a pretrained MLLM's vision encoder to replace traditional subword tokenization. All reviewers gave negative scores, with key concerns centered on limited novelty in contribution and unclear source of performance gains. Reject.

**Reviewer Concerns:**

Addressed by rebuttal:

- Similarity to VisInContext: Authors provided detailed differentiation in motivation (context extension vs. tokenization issues), research topic (multimodal vs. pure-text), and technical design (SEETOK eliminates text tokenization entirely).
- Cross-dataset validation: Added experiments on more benchmarks (TriviaQA, NQ, PopQA, SST5) confirming visual-text finetuning advantages.
- Long sequence scalability: Demonstrated 74k token handling with 23.6GB memory vs. baseline's 50k limit; added efficiency analysis in Appendix F.
- Ablation studies: Provided finer-grained ablations showing tuning vision encoder + LLM yields best results (43.53 on TriviaQA).
- Presentation (bold text): Removed excessive formatting per reviewer suggestion.
- Compositional analysis: Added cosine similarity analysis showing SEETOK incorporates both visual and semantic cues.

Outstanding concerns:

- Contribution novelty: Reviewer Kw1w's concern about incremental contribution over existing visual text approaches may not be fully resolved despite differentiation arguments.
- Source of Gain (QVXp):  A critical concern was whether the improvements stemmed from the SEETOK architecture or simply the instruction-tuning process. The authors' additional experiments in Table 4 show that after isolating the effects of instruction-tuning (comparing "Pure-Text + LoRA" to "SEETOK"), the net growth brought by SEETOK is not very obvious.

**Reviewer Scores:**

The paper received uniform initial scores of 4, 4, 4, and 4 from all reviewers. Throughout the evaluation process, no reviewer indicated that their primary concerns were sufficiently resolved to warrant an increase in their score.

---

### Decision · Program_Chairs · 2026-01-26

Reject